# PaCo: Parameter-Compositional Multi-Task Reinforcement Learning

**Lingfeng Sun**[1][*][†] **Haichao Zhang**[2][*] **Wei Xu**[2] **Masayoshi Tomizuka**[1]
[1]University of California Berkeley    [2]Horizon Robotics
lingfengsun@berkeley.edu  {haichao.zhang, wei.xu}@horizon.ai  tomizuka@berkeley.edu

## Abstract

The purpose of multi-task reinforcement learning (MTRL) is to train a single policy that can be applied to a set of different tasks. Sharing parameters allows us to take advantage of the similarities among tasks. However, the gaps between contents and difficulties of different tasks bring us challenges on both which tasks should share the parameters and what parameters should be shared, as well as the optimization challenges due to parameter sharing. In this work, we introduce a parameter-compositional approach (PaCo) as an attempt to address these challenges. In this framework, a policy subspace represented by a set of parameters is learned. Policies for all the single tasks lie in this subspace and can be composed by interpolating with the learned set. It allows not only flexible parameter sharing but also a natural way to improve training. We demonstrate the state-of-the-art performance on Meta-World benchmarks, verifying the effectiveness of the proposed approach.

## 1    Introduction

Deep reinforcement learning (RL) has made massive progress in solving complex tasks in different domains. Despite the success of RL in various robotic tasks, most of the improvements are restricted to single tasks in locomotion or manipulation. Although many similar tasks with different target and interacting objects are accomplished by the same robot, they are usually defined as individual tasks and solved separately. On the other hand, as intelligent agents, humans usually spend less time learning similar tasks and can acquire new skills using existing ones. This motivates us to think about the advantages of training a set of tasks with certain similarities together efficiently. Multi-task reinforcement learning (MTRL) aims to train an effective policy that can be applied to the same robot to solve different tasks. Compared to training each task separately, a multi-task policy should be efficient in the number of parameters and training samples and benefit from the sharing process.

The key challenge in multi-task RL methods is determining what should be shared among tasks and how to share. It is reasonable to assume the existence of similarities among all the tasks picked (usually on the same robot) since training completely different tasks together is meaningless. However, the gaps between different tasks can be significant even within the set. For tasks using the same skill but with different goals, it's natural to share all the parameters and add the goal into state representation to turn the policy into a goal-conditioned policy. For tasks with different skills, sharing policy parameters can be efficient for related tasks but may bring additional difficulties for uncorrelated skills (e.g., push and peg-insert-side in Meta-World [36]), due to additional challenges in learning brought by the conflicts between tasks.

Recent works on multi-task RL proposed different methods on this problem, which can be roughly divided into three categories. Some focus on modeling share-structures for sub-policies of different

---

[*]Equal contribution.
[†]Work done while interning at Horizon Robotics.

36th Conference on Neural Information Processing Systems (NeurIPS 2022).

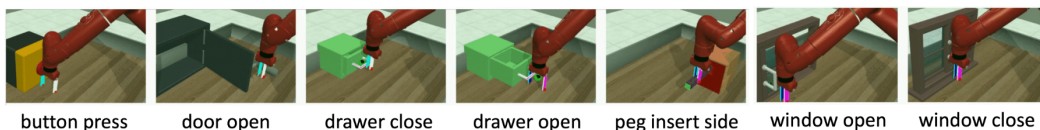

| button press | door open | drawer close | drawer open | peg insert side | window open | window close |

Figure 1: **Example tasks** from Meta-World [36].

tasks [3, 33], while some focus more on algorithms and aim to handle conflicting gradients from different tasks losses during training[35]. In addition, many works attempt to select or learn better representations as better task-condition for the policies [24]. In this paper, we focus on the share-structure design for multiple tasks. We propose a parameter-compositional MTRL method that learns a task-agnostic parameter set forming a subspace in the policy parameter space for all tasks. We infer the task-specific policy in this subspace using a compositional vector for each task. Instead of interpolating different policies' output in the action space, we directly compose the policies in the parameter space. In this way, two different tasks can have identical or independent policies. With different subspace dimensions (*i.e.*, size of parameter set) and additional constraints, this compositional formulation can unify many previous works on sharing structures of MTRL. Moreover, separating the task-specific and the task-agnostic parameter set brings advantages in dealing with instability RL training of certain tasks, which helps improve the multi-task training stability.

The key contributions of our work are summarized as below. *i)* We present a general Parameter Compositional (PaCo) MTRL training framework that can learn representative parameter sets used to compose policies for different tasks. *ii)* We introduce a scheme to stabilize MTRL training by leveraging PaCo's decompositional structure. *iii)* We validate the state-of-the-art performance of PaCo on Meta-World benchmark compared with a number of existing methods.

## 2 Preliminaries

### 2.1 Markov Decision Process (MDP)

A discrete-time Markov decision process is defined by a tuple $(\mathcal{S}, \mathcal{A}, P, r, \mu, \gamma)$, where $\mathcal{S}$ is the state space; $\mathcal{A}$ is the action space; $P$ is the transition process between states; $r : \mathcal{S} \times \mathcal{A} \rightarrow \mathbb{R}$ is the reward function; $\mu \in \mathcal{P}(\mathcal{S})$ is distribution of the initial state, and $\gamma \in [0, 1]$ is the discount factor. At each time step $t$, the learning agent generates the action with a policy $\pi(a_t|s_t)$ as the decision. The goal is to learn a policy to maximize the accumulated discounted return.

### 2.2 Soft Actor-Critic

In the scope of this work, we will use Soft Actor-Critic (SAC) [9] to train the universal policy for the multi-task RL problem. SAC is an off-policy actor-critic method that uses the maximum entropy framework. The parameters in SAC framework include the policy network $\pi(a_t|s_t)$ used in evaluation, the critic network $Q(s_t, a_t)$ as a soft Q-function. A temperature parameter $\alpha$ is used to maintain the entropy level of policy. In multi-task learning, the one-hot id of task skill and the goal is appended to the state space. Different from single-task SAC, multiple tasks may have different learning dynamics. Therefore, we follow previous works [24] to assign a separate temperature $\alpha_\tau$ for each task with different skills. The policy and critic function optimization procedure remains the same as the single-task setting.

## 3 Revisiting and Analyzing Multi-Task Reinforcement Learning

### 3.1 Multi-Task Reinforcement Learning Setting

Each single task can be defined by a unique MDP, and changes in state space, action space, transition, reward function can result in completely different tasks. In MTRL, instead of solving a single MDP, we solve a bunch of MDPs from a task family using a universal policy $\pi_\theta(a|s)$. The first assumption for MTRL is to have a universal shared state space $\mathcal{S}$ and each task has a disjoint state space $S^\tau \subset S$, where $\tau \in \mathcal{T}$ is any task from the full task distribution. In this way, the policy would be able to

recognize which task it is currently solving. Adding the one-hot encoding for task id is a common implementation of getting disjoint state space during experiments.

In general MTRL setting, we don't have strict restrictions on the which tasks are involved, but we assume that tasks in the full task distribution share some similarities. In real applications, depending on how a task is defined, we can divide it into Multi-Goal MTRL and Multi-Skill MTRL. For the former one, the task set is defined by various "goals" in the same environment. The reward function $r^\tau$ is different for each goal, but the state and transition remains the same. Typical examples of this Multi-goal settings are locomotion tasks like *Cheetah-Velocity/Direction* [3] and all kinds of goal-conditioned manipulation tasks [18]. For the later one, besides changes in goals in the same environment, the task set also involves different environments that share similar dynamics (transition functions). This happens more in manipulation tasks where different environments train different skills of a robot, and one natural example is the Meta-World [36] benchmark which includes multiple goal-conditioned manipulation tasks using the same robot arm. In this setting, the state space of different tasks changes across different skills since the robot is manipulating different objects (*c.f.* Figure 1). In both Multi-goal and Multi-skill setting, we have to form the set of MDPs into a universal Multi-task MDP and find a universal policy that works for all tasks. For multi-goal tasks, we need to append "goal" information into state; for multi-skill tasks, we need to append "goal" (usually position) as well as "skill" (usually one-hot encoding). After getting state $\mathcal{S}^\tau$, the corresponding transition and reward $P^\tau, r^\tau$ can be defined accordingly.

### 3.2 Challenges in Multi-Task Reinforcement Learning

**Parameter-Sharing.** Multi-task learning aims to learn a single model that can be applied to a set of different tasks. Sharing parameters allows us to take advantage of the similarities among tasks. However, the gaps between contents and difficulties of different tasks bring us the challenges on both which tasks should share the parameters and what parameters should be shared. Failure in the design may result in low success rate on certain tasks that could have been solved if trained separately. *This is a challenge in designing an effective structure to solve the MTRL task.*

**Multi-Task Training Stability.** Although we have assumed some similarity in the task sets used for multi-task learning, conflicts between different skills may affect the whole training process [35]. Also, failure like loss explosion in some tasks can severely affect the training of other tasks due to parameter sharing [24]. In multi-task training with large task numbers, the uncertainty of single task training is enlarged. *This is a challenge in designing an algorithm to avoid negative influence brought by parameter-sharing among multiple tasks.*

## 4 Parameter-Compositional Multi-Task RL

Motivated by the challenges in training universal policies for multiple tasks discussed in Section 3, we will present a Parameter-Compositional approach to MTRL. The proposed approach is conceptually simple, yet offers opportunities in addressing the MTRL challenges as detailed in the sequel.

### 4.1 Formulation

In this section, we describe how we formulate the parameter-compositional framework for MTRL. Given a task $\tau \sim \mathcal{T}$, where $\mathcal{T}$ denotes the set of tasks with $|\mathcal{T}| = T$, we use $\boldsymbol{\theta}_\tau \in \mathbb{R}^n$ to denote the vector of *all the trainable parameters* of the model (*i.e.*, policy and critic networks) for task $\tau$. We employ the following decomposition for the *task parameter vector* $\boldsymbol{\theta}_\tau$:

$$\boldsymbol{\theta}_\tau = \boldsymbol{\Phi}\mathbf{w}_\tau, \tag{1}$$

where $\boldsymbol{\Phi} = [\boldsymbol{\phi}_1, \boldsymbol{\phi}_2, \cdots, \boldsymbol{\phi}_i, \cdots, \boldsymbol{\phi}_K] \in \mathbb{R}^{n \times K}$ denotes a matrix formed by a set of $K$ parameter vectors $\{\boldsymbol{\phi}_i\}_{i=1}^K$ (referred to as *parameter set*, which is also overloaded for referring to $\boldsymbol{\Phi}$), each of which has the same dimensionality as $\boldsymbol{\theta}_\tau$, *i.e.*, $\boldsymbol{\phi}_i \in \mathbb{R}^n$. $\mathbf{w}_\tau \in \mathbb{R}^K$ is a *compositional vector*, which is implemented as a trainable embedding vector for the task index $\tau$. We refer a model with parameters in the form of Eqn.(1) as a *parameter-compositional* model.

---

[3] By *Cheetah/Ant-Velocity/Direction*, we refer to the tasks that have the same dynamics as the standard locomotion tasks but with a goal of running at a specific velocity or in a specific direction.

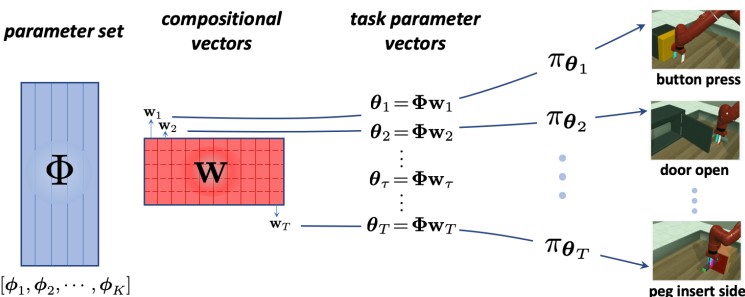

Figure 2: **Parameter-Compositional method (PaCo)** for multi-task reinforcement learning. In this framework, the network parameter vector $\boldsymbol{\theta}_\tau$ for a task $\tau$ is instantiated in a compositional form based on the *shared* base parameter set $\boldsymbol{\Phi}$ and the *task-specific* compositional vector $\mathbf{w}_\tau$. Then the networks are used in the standard way for generating actions or computing the loss [9]. During training, $\boldsymbol{\Phi}$ will be impacted by all the task losses, while $\mathbf{w}_\tau$ is impacted by the corresponding task loss only.

In the presence of a single task, the decomposition in Eqn.(1) brings no additional benefits, as it is essentially equivalent to the standard way of parameterizing the model. However, when faced with multiple tasks, as in the MTRL setting considered in this work, the decomposition in Eqn.(1) offers opportunities for tackling the challenges posed by the MTRL setting. More concretely, since Eqn.(1) decomposes the parameters to two parts: *i)* task-agnostic $\boldsymbol{\Phi}$ and *ii)* task-aware $\mathbf{w}_\tau$, we can share the task-agnostic $\boldsymbol{\Phi}$ across all the tasks, while still ensure task awareness via $\mathbf{w}_\tau$, leading to:

$$[\boldsymbol{\theta}_1, \cdots, \boldsymbol{\theta}_\tau, \cdots, \boldsymbol{\theta}_T] = \boldsymbol{\Phi}[\mathbf{w}_1, \cdots, \mathbf{w}_\tau, \cdots \mathbf{w}_T]$$
$$\boldsymbol{\Theta} = \boldsymbol{\Phi}\mathbf{W}. \tag{2}$$

For MTRL, let $J_\tau(\boldsymbol{\theta})$ denotes the summation of both actor and critic losses implemented in the same way as in SAC [9] for task $\tau$, the multi-task loss is defined as the summation of individual loss $J_\tau$ across tasks:

$$J_{\boldsymbol{\Theta}} \triangleq \sum_\tau J_\tau(\boldsymbol{\theta})$$

where $\boldsymbol{\Theta}$ denotes the collection of all the trainable parameters of both actor and critic networks. Together with Eqn.(2), it can be observed that the multi-task loss $J_{\boldsymbol{\Theta}}$ contributes to the learning of the model parameters in two ways:

- $\partial J_{\boldsymbol{\Theta}} / \partial \boldsymbol{\Phi} = \sum_\tau \partial J_\tau / \partial \boldsymbol{\Phi}$: all the $T$ tasks contribute to the learning of the shared parameter set $\boldsymbol{\Phi}$;
- $\partial J_{\boldsymbol{\Theta}} / \partial \mathbf{W} = \sum_\tau \partial J_\tau / \partial \mathbf{w}_\tau$: as for the training of the the task specific compositional vectors, each task loss $J_\tau$ will impact only its own task specific compositional vector $\mathbf{w}_\tau$.

The PaCo framework is illustrated in Figure 2. Additional implementation details about PaCo are provided in Appendix A.2. The proposed approach has several attractive properties towards addressing the MTRL challenges discussed earlier:

- the compositional form of parameters as in Eqn.(2) offers flexible parameter sharing between tasks, by learning the appropriate compositional vectors for each task over the shared parameter set;
- because of the clear separation between task-specific and task-agnostic parameters, it also offers a natural solution for improving the stability of MTRL training, as detailed in the sequel.

In addition, the separation between task-specific and task-agnostic information has other benefits that beyond the scope of the current work. For example, the task-agnostic parameter set $\boldsymbol{\Phi}$ could be reused as a pre-trained policy basis in some transfer scenarios (initial attempts in Appendix A.3).

## 4.2 Stable Multi-Task Reinforcement Learning

One inherent challenge in MTRL is the interference during training among tasks due to parameter sharing. One consequence of this is that the failure of training on one task may adversely impact the training of other tasks [28, 35]. For example, it has been empirically observed that some task losses may explode during training on Meta-World [24], which will contribute a significant portion in

---

**Algorithm 1** Parameter-Compositional MTRL (PaCo)[5]

---

**Input:** param-set size $K$, loss threshold $\epsilon$, learning rate $\lambda$
**while** termination condition is not satisfied **do**
    $\boldsymbol{\theta}_\tau = \boldsymbol{\Phi}\mathbf{w}_\tau$         ▷ compose task parameter vector
    $J_\tau \leftarrow J_\tau(\boldsymbol{\theta})$        ▷ loss (actor+critic as in SAC) across tasks
    (*Reset Step 1: loss maskout*)  $J_\eta \leftarrow 0$   if $J_\eta > \epsilon$
    $J_\boldsymbol{\Theta} \leftarrow \sum_\tau J_\tau$      ▷ calculate multi-task loss
    $\boldsymbol{\Phi} \leftarrow \boldsymbol{\Phi} - \lambda \nabla_{\boldsymbol{\Phi}} J_\boldsymbol{\Theta}$     ▷ parameter set update
    **for** each task $\tau$ **do**
        $\mathbf{w}_\tau \leftarrow \mathbf{w}_\tau - \lambda \nabla_{\mathbf{w}_\tau} J_\tau(\mathbf{w}_\tau)$   ▷ composition parameter update
    **end for**
    (*Reset Step 2:* **w**-*reset*)  $\mathbf{w}_\eta \leftarrow$ Eqn.(3)   if $J_\eta > \epsilon$
**end while**

---

updating the shared parameters because of their dominance. As a consequence, this will significantly impact the training of the other tasks through the shared parameters. To mitigate this issue, [24] adopted an empirical trick by terminating the training once this issue is spotted and the whole training is discarded (referred to as *stop-relaunch*). [4] Here we show that in PaCo, there is a natural way to mitigate this issue without resorting to an ad-hoc training pipeline [24] or more expensive schemes [35].

The straightforward idea is to mask out $J_\eta$ of task $\eta$ with exploding loss from the total loss $J$ to avoid its adverse impacts on others. More specifically, once a task loss $J_\eta$ surpasses some threshold $\epsilon$, it will be excluded from the training loss. We will refer to this step as *loss maskout*.

Because of the compositional nature of the PaCo model, we can introduce further improvement beyond loss maskout. This can be achieved by re-initializing $\mathbf{w}_\eta$ without impacting the parameters of other tasks and then following the normal training. One way to re-initialize $\mathbf{w}_\eta$ is:

$$\mathbf{w}_\eta = \sum_{j \in \mathcal{V}} \beta_j \mathbf{w}_j, \quad \boldsymbol{\beta} = [\beta_1, \beta_2, \cdots] \sim \Delta^{|\mathcal{V}|-1} \tag{3}$$

where $\mathcal{V} \triangleq \{j | J_j \leq \epsilon\}$, and $\boldsymbol{\beta}$ is uniformly sampled from a unit $|\mathcal{V}|-1$-simplex $\Delta^{|\mathcal{V}|-1}$. We refer to this step as **w**-*reset*. During training, *w-reset* offers an opportunity to continue learning for the task with exploding loss by resetting its $w$-parameter, which will generate a new task parameter vector when composed with $\boldsymbol{\Phi}$ and $w_\eta$. At the same time, this reset has no influence on other non-exploding tasks, which provides opportunities to further improves over *loss maskout*. The overall stabilization scheme including both *loss maskout* and **w**-*reset* is termed as *Reset*. The complete procedure of PaCo is presented in Algorithm 1.

It is worthwhile to point out that the ability to use the scheme of **w**-*reset* is a unique feature of PaCo, due to its clear separation between task-agnostic and task-specific parameters. Previous methods such as Soft Modularization [33] and CARE [24] cannot employ this due to the lack of clear decomposition between the two parts. Instead, the *stop-relaunch* trick is applied to all the baselines following [24], which is related to but more expensive than loss maskout as it discards the whole training. Results show that the proposed *Reset* scheme can improve training with better performance (*c.f.* Table 1∼2).

### 4.3 Unified Perspective on Some Existing Methods

Apart from the interesting compositional form and the features of PaCo, it also provides a unified perspective on viewing some existing methods. Using this formulation, we are able to re-derive some existing methods with specific instantiations of $\boldsymbol{\Phi}$ and $\mathbf{w}$.

- **Single-Task Model**: if set $\boldsymbol{\Phi} = [\boldsymbol{\phi}_1, \boldsymbol{\phi}_2 \cdots]$ and $\mathbf{w}_\tau$ as a one-hot task-id vector, this essentially instantiates a single-task model, *i.e.* each task has its dedicated parameters.
- **Multi-Task Model**: if we set $\boldsymbol{\Phi} = [\boldsymbol{\phi}] \in \mathbb{R}^{n \times 1}$, $\mathbf{w}_1 = \mathbf{w}_2 = ... = 1$, then all the tasks share the same parameter vector $\theta^\tau = \phi$. By taking state and the task-id as input, we have the multi-task model.

---

[4] https://github.com/facebookresearch/mtrl/blob/eea3c99cc116e0fadc41815d0e7823349fcc0bf4/mtrl/agent/sac.py#L322
[5] To highlight the core algorithm, we have omitted the steps that are identical to standard SAC [9], including environmental unroll, temperature tuning and target critic update.

- **Multi-Head Multi-Task Model**: by setting $\mathbf{\Phi}$ as follows:

$$\mathbf{\Phi} = \begin{bmatrix} \boldsymbol{\phi}' & \boldsymbol{\phi}' & \cdots & \boldsymbol{\phi}' & \cdots & \boldsymbol{\phi}' \\ \boldsymbol{\psi}_1 & \boldsymbol{\psi}_2 & \cdots & \boldsymbol{\psi}_\tau & \cdots & \boldsymbol{\psi}_K \end{bmatrix} \in \mathbb{R}^{n \times K}$$

where $\boldsymbol{\psi}_\tau$ is the sub-parameter-vector of the output layer for task $\tau$. Setting $\mathbf{w}_\tau$ as a one-hot task-id vector, we recover the multi-head model for MTRL, where all the tasks share the same trunk network parameterized by $\boldsymbol{\phi}'$ with independent head $\boldsymbol{\psi}^\tau$ for each task $\tau$.

- **Soft-Modularization** [33] divides each layer into several groups of "modules" ($\{f_{\boldsymbol{\phi}_j^i}\}$) and then combines their outputs with "soft weights" $\boldsymbol{z}(s, \tau)$ from another "routing" network. To obtain these soft weights, the routing network takes both the task id and state as input. Mathematically, it can be represented as

$$f_{\boldsymbol{\theta}_\tau} = \begin{bmatrix} [f_{\boldsymbol{\phi}_1^1} & f_{\boldsymbol{\phi}_2^1} & \cdots & f_{\boldsymbol{\phi}_K^1}]\boldsymbol{z}^1(s, \tau) \\ & & \vdots & \\ [f_{\boldsymbol{\phi}_1^m} & f_{\boldsymbol{\phi}_2^m} & \cdots & f_{\boldsymbol{\phi}_K^m}]\boldsymbol{z}^m(s, \tau) \end{bmatrix},$$

where $\mathbf{\Phi}$ is in a specially structured form, with the combination done at each level with a "per-level" soft combination vector $\boldsymbol{z}(s, \tau)$ conditioned on current state $s$ and task-id $\tau$.

The dependency of the combination vector $\boldsymbol{z}(s, \tau)$ on state $s$ makes it diffuse task-relevant and task-agnostic information together; therefore, all the parameters are entangled with state information and are less flexible in some use cases. For example, $\mathbf{w}$-*reset*-like operation is inapplicable to Soft-Modularization [33] because of the mixed role of $\boldsymbol{z}$ on state $s$ and task $\tau$.

## 5 Related Work

**Multi-Task Learning.** Multi-Task learning is one of the classical paradigm for learning in the presence of multiple potentially related tasks [4]. It holds the promise that the joint learning of multiple tasks with a proper way of information sharing can make the learning effective. It has been extensively investigated from different perspectives [12, 14, 21, 22, 38, 27, 1, 15] and been applied in many different fields, including computer vision [37, 13, 16, 26], natural language processing [31, 25] and robotics [11, 2].

**Multi-Task Reinforcement Learning.** The idea of multi-task learning has also been explored in MTRL, with a similar objective of improving the performance of single-task RL by exploiting the similarities between different tasks. Many different approaches have been proposed in the literature [3, 5, 10, 8, 36, 17, 33, 24, 32, 20]. One of the most straight-forward approach to MTRL is to formulate the multi-task model as a task-conditional one [36], as commonly used in goal-conditional RL [18] and visual-language grounding [17]. Although simple and has shown some success in certain cases, one inherent limitation is that it is more vulnerable to the negative interferences among tasks, because of the complete sharing of network parameters. [3] proposes an approach by assuming the functional approximator for each task is linear combination of a set of shared feature vectors, and then exploited the similarities among different tasks by employing a structured sparse penalty over the combination matrix. [8] utilizes a mix-and-match design of the model to facilitate transferring between tasks and robots. [7] leverages the shared knowledge between multiple tasks by using a shared network followed by multiple task-specific heads. [33] further extends these approaches by softly sharing features (activations) from a base network among tasks, by generating the combination weight with an additional modularization network taking both state and task-id as input. Since the base and modularization networks take state and task information as input, there is no clear separation between task-agnostic and task-specific parts. This limits its potential on tasks such as continual learning of a novel task. Differently, PaCo explores a compositional structure in the *parameter space* [19, 30] instead of in feature/activation space [3, 33], and does so in a way such that the task-agnostic and task-specific parts are decomposed. This not only enables learning of multiple tasks, but also leads to a natural schemes for stabilizing and improving MTRL training (*c.f.* Sec.6.2).

**Resolving Conflicts in Multi-Task Learning.** Because of parameter sharing for multiple tasks (thus multiple task losses), the shared parameters are impacted by the gradients from all the task losses. Whenever the gradients is not consistent with each other, there will be conflicts in updating the shared parameter. This issue of conflicting gradients is a general problem that is present in general multi-task learning [35, 29], and could lead to degraded performance and unstable training

if not properly handled [24]. [28] bypassed the conflicting gradient issue by discarding parameter sharing, but instead distilling each task policy into a centralized policy. [24] alleviates the negative effects of interference by deciding which information should be shared across tasks, using a context-based attention over a mixture of state encoders. This demonstrates the benefits of a task-grouping mechanism but requires additional context information. There are also approaches on mitigating the interferences by balancing of the multiple tasks from the perspective of loss [10] or gradient [6]. [35] proposes to address the conflicts by gradient projection, which could be less reliable in the case where gradients are noisy, as is the case in RL. Differently, PaCo enjoys improved stability in training by using schemes leveraging its decomposed structure of task-agnostic and task-specific parameters.

## 6   Experiments

We now empirically test the performance of our Parameter-Compositional Multi-Task RL framework on the Meta-World benchmark [36]. Meta-World benchmark is a robotic environment consisting a number of distinct manipulation tasks, with some examples tasks shown in Figure 1. Each task itself is a goal-conditioned environment, and the state space of all the tasks has the same dimension. The action space of different task is exactly the same, but certain dimensions in the state space represent different semantic meaning in different tasks (*e.g.* goal position or object position).

### 6.1   Benchmark Results on Meta-World

**Benchmarks.** Meta-World [36] has been used in benchmarking many recent MTRL algorithms [33, 24]. In the original Meta-World Multi-task benchmark [36], each manipulation task is configured with a fixed goal, therefore the learned policies are not goal-conditioned as it cannot generalize to a task of the same type by with different goals. This setting is easier, but more restrictive and less realistic in robotic learning [33]. Following [33], we extend all the tasks to a random-goal setting and refer to the 10 task Meta-World with random goals as MT10-rand.

**Baselines.** We compare against *(i)* **Multi-task SAC**: extended SAC [9] for MTRL with one-hot task encoding; *(ii)* **Multi-Head SAC**: SAC with shared a network apart from the output heads, which are independent for each task; *(iii)* **SAC+FiLM**: the task-conditional policy is implemented with the FiLM module [17] on top of SAC; *(iv)* **PCGrad** [35]: a representative method for handling conflicting gradients during multi-task learning via gradient projection during optimization; *(v)* **Soft-Module** [33]: which learns a routing network that guides the soft combination of modules (activations) for each task; *(vi)* **CARE** [24]: a recent method that achieves the state-of-the-art performance on Meta-World benchmark by leveraging additional task-relevant metadata for state representation.[6]

**Training Settings.** The convergence performance is related to both the number of parallel environments for training and the number of tasks in the environments. There is also a balance between the number of training iterations and the number of roll-out samples. For training on MT10-rand, we follow the settings introduced in [24] and use *i)* 10 parallel environments, *ii)* 20 million environment steps for the 10 tasks together (2 million per task), *iii)* repeated training with 10 different random seeds for each method. The implementation of PaCo and the training scripts are available. [7] More resources are available on the project page. [8]

**Evaluation Metrics and Results.** The evaluation metric for the learned universal policy for all tasks is based on the success rate of the policy for all the tasks. For Meta-World benchmarks, we evaluate each skill with 10 episodes of different sampled goals using the final policy. The success rate is then averaged across all the skills. The randomness in the MTRL training is unpredictable, some methods may converge to higher success rate right after 20M steps, and some may drop if the training continues.

| Methods | Success Rate (%) (mean ± std) |
|---|---|
| Multi-Task SAC [36] | 62.9 ± 8.0 |
| Multi-Head SAC [36] | 62.0 ± 8.2 |
| SAC + FiLM [17] | 58.3 ± 4.3 |
| PCGrad [35] | 61.7 ± 10.9 |
| Soft-Module [33] | 63.0 ± 4.2 |
| CARE [24] | 76.0 ± 6.9 |
| **PaCo** (Ours) | **85.4 ± 4.5** |

Table 1: Results on Meta-World [36] MT10 with random goals (MT10-rand).

---

[6]Note that the experiments reported in the works mentioned above are implemented and evaluated on Meta-World-V1 and/or with the fixed-goal setting. We adapt these methods and experiment on Meta-World-V2.

[7]`https://github.com/TToTMooN/paco-mtrl`

[8]`https://sites.google.com/site/hczhang1/projects/paco-mtrl`

Instead of picking the maximum evaluation success rate across training, we use the policy at 20M total environment steps (2M per task) for fair evaluation. We report the mean performance together with standard derivations of the models trained with 10 different seeds, as summarized in Table 1. An improved success rate on MT10-rand benchmark can be observed compared to baseline methods. It is interesting to note that the previous state-of-the-art method CARE [24] leverages additional metadata such as language-based task-description for helping with learning. PaCo outperforms CARE without resorting to such meta information, highlighting the benefits brought by algorithmic innovation itself. Note that all the baseline results are obtained with the empirical trick of terminating and re-launching training in the presence of exploding loss [24]. More results on MT10 and MT50 are provided in Appendix A.1.2.

**Single-Task Performance.** The tasks in Meta-World are designed such that they can be trained with standard single-task RL algorithms (e.g. SAC). Given enough environmental steps (varies for different tasks), they can converge to close to a 1.0 success rate. The average success rate at convergence for the 10 tasks in MT10 is around 95.0% (each SAC policy is trained with 5 Million environmental steps per task). In some previous works [24], performance under this setting is regarded as an "upper bound". As a reference, a 2M steps per task setting for Single-Task SAC training results in a 61.9% average success rate. Single-Task SAC cannot reach its convergence performance for all tasks if limited to 2M per task for training.

## 6.2 Stable MTRL Training

In single task RL, training failure or gradient explosion may occur due to many random factors including initialization, bad exploration *etc.*. This is even worse for MTRL, especially on those models with shared parameters, since the loss and gradient explosion on certain tasks would influence other tasks through the shared parameters. In PaCo, we use the loss maskout and $\mathbf{w}$-reset schemes introduced in Section 4.2 to avoid the influence of extreme losses of certain tasks. We perform an ablation study on them to analyze their roles in stabilizing the MTRL training. We compare PaCo with two variations: *i*) PaCo-*Maskout*: the PaCo variant with only loss maskout, and *ii*) PaCo-*Vanilla*: PaCo *without* $\mathbf{w}$-reset and loss maskout.

| Variations | Stabilization Scheme | | Success Rate (%) |
| --- | --- | --- | --- |
| | loss maskout | $\mathbf{w}$-reset | |
| PaCo | ✓ | ✓ | $85.4 \pm 4.5$ |
| PaCo-*Maskout* | ✓ | ✗ | $76.0 \pm 8.8$ |
| PaCo-*Vanilla* | ✗ | ✗ | $71.6 \pm 25.7$ |

Table 2: Stable MTRL training results on MT10-rand tasks.

In order to show the stability precisely, we don't early-stop the training process even if the loss of some task has already exploded. The final success rates are shown in Table 2 (training curves provided in Appendix A.1.1). It is observed that PaCo-*Vanilla* has a larger variation and is less stable during training (*c.f.* training curves in Appendix A.1.1), resulting from exploding loss in some cases. PaCo-*Maskout* variant can potentially mitigate the adverse effects of the exploding loss to some degree, but could also compromise the performance because of the reduced opportunity of learning on the masked-out tasks. The complete PaCo improves both the average success rate and the stability of MTRL training compared with both variants, demonstrating the effectiveness of the PaCo design. More details on stability comparison in the training process can be found in Appendix A.1.1.

## 6.3 Ablation Study

**Parameter Set Size.** One goal of MTRL is to find a good trade-off between model size and performance. The hyper-parameter in PaCo for controlling this trade-off is the size of parameter set $K$. For a group of similar skills, we may be able to reach high success rate with a small parameter set. For skills with high variance, a larger parameter set might be required due to the increased diversity.

| Param Set Size | Success Rate (%) | | Compositional Variations | Success Rate (%) |
| --- | --- | --- | --- | --- |
| PaCo (K=3) | $74.4 \pm 5.1$ | | Output-only | $64.0 \pm 5.5$ |
| **PaCo** (K=5) | $85.4 \pm 4.5$ | | Actor-only | $73.3 \pm 5.8$ |
| PaCo (K=8) | $81.0 \pm 8.4$ | | AC-shared (**PaCo**) | $85.4 \pm 4.5$ |

Table 3: Impacts of (a) parameter set size and (b) compositional structure variations.

We investigated the performance of PaCo under different values for the parameter set size $K$. More concretely, for the benchmark MT10-rand which contains 10 different manipulation skills, we set

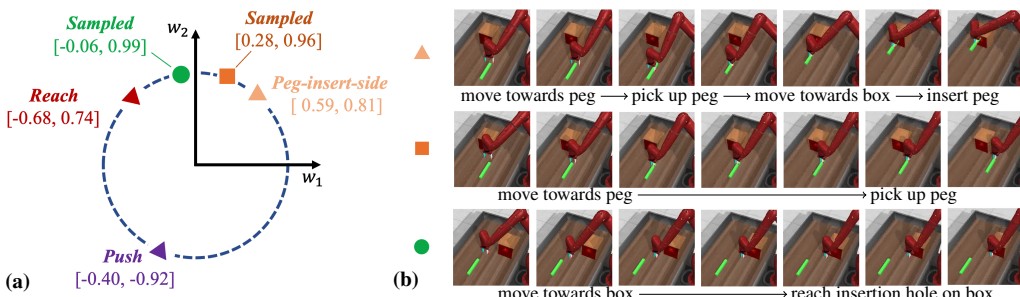

Figure 3: **(a)** Compositional vectors in the space of a unit circle: *learned* reach, push, peg-inset-side policies (denoted with △) and *sampled* policies on the unit circle. **(b)** Visualization (using the *peg-insert-side* task) of example policies: the *learned* peg-inset-side policy and two *sampled* ones.

$K = 3, 5, 8$, and the results are summarized in Table 3 (a). It is observed that with a small value for $K$, PaCo can already achieve performance competitive to many baseline methods (*c.f.* Table 1). Naively increasing $K$ to a large number could compromise the performance, partially due to decreased sample efficiency because of the additional complexities from over-parameterization. However, in a special case of $K = 10$, by initializing $\mathbf{w}$ as one-hot vectors, PaCo can achieve $88.9 \pm 1.9$. We leave further improvements based on better initialization in the general case to future work. Empirically we find setting $K = 5$ lead to good trade-off for PaCo. Overall, the increasing in parameter set size $K$ grows with the task number $n$ but essentially at a lower speed.

**Compositional Structure Variations.** PaCo employs the same compositional structure to all (actor and critic) networks (*AC-shared*). There are several other possible variations on this design choice. One variant is to use the compositional structure only for the parameters of the actor-network (*Actor-only*). Also, we can choose to apply the structure only to the output layer (*Output-only*), which is architecturally similar to Multi-Head SAC. The results in Table 3(b) show that when applied to output layer only, the performance is comparable to Multi-Head SAC (*c.f.* Table 1), although with less parameters ($K = 5$) than Multi-Head SAC ($K = 10$). This shows that the compositional structure is also effective in the output layer up to the performance limit imposed by the network architecture. Using the compositional structure for the full actor network (*Actor-only*) improves over *Output-only*. Applying the compositional structure to the full networks of both actor and critic can further unleash its potential and gives the best performance.

### 6.4 Qualitative Results on Parameter Set and Compositional Vectors

**Visualization of Samples from Policy Subspace.** We conduct experiments to inspect the policy subspace spanned by the learned parameter set $\mathbf{\Phi}$ via visualization. For ease of visualization, we train PaCo on a representative set of 3 tasks (*reach*, *push* and *peg-insert-side* with random goals), using a parameter set size of $K = 2$, *i.e.*, $\mathbf{\Phi} = [\phi_1, \phi_2]$ and $\mathbf{W} = [\mathbf{w}_1, \mathbf{w}_2, \mathbf{w}_3] \in \mathbb{R}^{2 \times 3}$. The three tasks cover simple, medium, and hard tasks, using the number of environmental interactions required to solve the task as the criteria for the level of difficulty. To better visualize the subspace, we further add a *normalize* activation to the compositional vectors, *i.e.*, all the policies lie on the unit circle in the $W$ space (the space of all possible $\mathbf{w}$ vectors, *c.f.* Figure 3 (a)). Although not the main focus of this experiment, as a side note, for this group of tasks, it is typically difficult to learn using one model like MT-SAC or Multi-Head SAC, with a low success rate for the most difficult task (*peg-insert-side*). In contrast, PaCo reaches a success rate close to $100\%$ across all three tasks.

In Figure 3 (a), we show the position of $\{\mathbf{w}_i\}$ for three task-specific policies on the unit circle. It can be observed that: *i)* We are not learning all the tasks with a single column of $\mathbf{\Phi}$. Instead, all columns of $\mathbf{\Phi}$ are involved in solving the three tasks with different compositional weights. *ii)* We are able to find policies for three completely different manipulation skills on this 1D low-dimensional manifold in the policy parameter space. This aligns well with the purpose of multi-task learning in sharing parameters. *iii)* The learned two column of parameter set ($[\phi_1, \phi_2]$) can be viewed as "basis" policies. Although individually, they may not necessarily correspond to a fully capable skill for a particular task, collectively, they can be used to instantiate various other policies through parameter space composition, including the policies for solving the three example tasks (*reach*, *push* and *peg-insert-side*). Since $[\phi_1, \phi_2]$ are not random vectors but have already been trained on multiple tasks, the policies instantiated with a random $\tilde{\mathbf{w}}$ could lead to policies with different types of behaviors,

possibly with some interactions with objects in the environment. As a test, we sampled some points on the unit circle (labeled as *Sampled*). The resulting policies are visualized in Figure 3 (b), showing various behaviors with some potential interactions with objects in the *peg-insert-side* environment.

**Visualization of Compositional Vectors.** We visualize the learned compositional vectors for MT10-rand in a 2D space (referred to as **w**-space) via Principal Component Analysis (PCA) in Figure 4.

There are several interesting observations: *i)* Some skills could lie in different part of the **w**-space, *e.g. reach v.s.* others; *ii)* Some skills are close in the **w**-space, *e.g. window-open v.s door-open*, *window-open v.s. window-close* in Figure 4; *iii)* Another interesting observation is that some skills that are not literally related are also appear to be close in the **w**-space learned by PaCo, *e.g. peg-insert v.s. window-open/window-close/door-open/drawer-open*. Although literally distinct, *peg-insert* and the other skills mentioned above are related from the perspective of behavior, *i.e.*, first interacting with an object (*e.g., peg/window/door*), and then taking a trajectory of motions to accomplish the subsequent operation, (*e.g. insert/window/door*). Therefore, these literally unrelated skills appear to be semantically related at the behavior

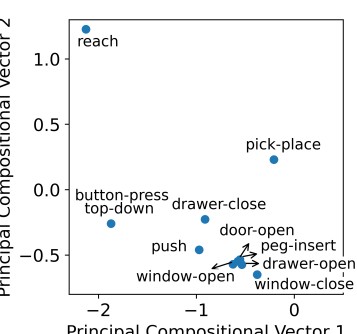

Figure 4: 2D PCA projection of the compositional vectors.

level. This is something that could be useful but is not able to be leveraged by CARE [24], implying one possible reason why PaCo is more effective. The learned compositional vectors as well as more visualization results can be found in Appendix A.1.4.

## 7 Conclusion, Limitation and Future Work

Based on a revisit to MTRL and its challenges, we present PaCo, a simple parameter compositional approach as a way to mitigates some of these challenges. The proposed approach has the benefits of clear separation between task-agnostic and task-specific components, which is not only flexible in learning but also useful for stabilizing and improving MTRL. Without resorting to more complicated design [33] or additional metadata [24], PaCo has demonstrated clear improvement over current state-of-the-art methods on standard benchmarks.

One limitation of the proposed approach is that a simple linear compositional form is used globally, which may limit its representation power in some cases. How to extend the current method beyond simple global linear compositional form while retaining its key advantages is one interesting direction to explore in the future.

There are several other interesting directions that are worth of exploration as well (with some initial attempts in Appendix A.3). For example, incorporating a higher-level mechanism for adjusting the task distributions based on task difficulties or progresses [34] to further improve the training of the more difficult tasks. The separation between task-agnostic and task-relevant parameters in PaCo also suggests a possibility for extending it to transfer learning.

## Acknowledgement

We would like to thank group members at Horizon Robotics and anonymous reviewers for discussions and feedback on the project and paper. We would also like to thank the Horizon AI platform team for infrastructure support.

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
