# A   Appendix

In appendix, we provide some additional results in Section A.1, more implementation details in Section A.2 and some initial attempts on some possible future extensions of PaCo in Section A.3.

## A.1   Additional Results

### A.1.1   Analyze Stability of PaCo on MT10-rand from Training Curve

In Figure 5, we show the evaluated average success rate of the three variations (PaCo, PaCo-*Maskout*, PaCo-*Vanilla*) in the MT10-rand experiments. To compare the stability of training, we didn't early-stop the training process even if the loss of some tasks already exploded. PaCo-*Vanilla* has a larger variation and is less stable during training due to the exploding loss of some tasks. The variance is very large compared to other methods since it can reach high performance (0.90 at 20M steps) for some random seeds but can perform very poorly (0.28 at 20M steps) for some other seeds. PaCo-*Maskout* variant is more stable compared to the vanilla version with masked-out extreme loss and can mitigate the adverse effects of the exploding loss of some tasks to other tasks. However, it can also compromise the performance because of the reduced opportunity of learning some tasks once they are masked out. The complete PaCo improves both the average success rate and the stability of MTRL training compared with both variants, demonstrating the effectiveness of the PaCo design. It reaches at least a 0.8 average success rate for all the random seeds used in experiments.

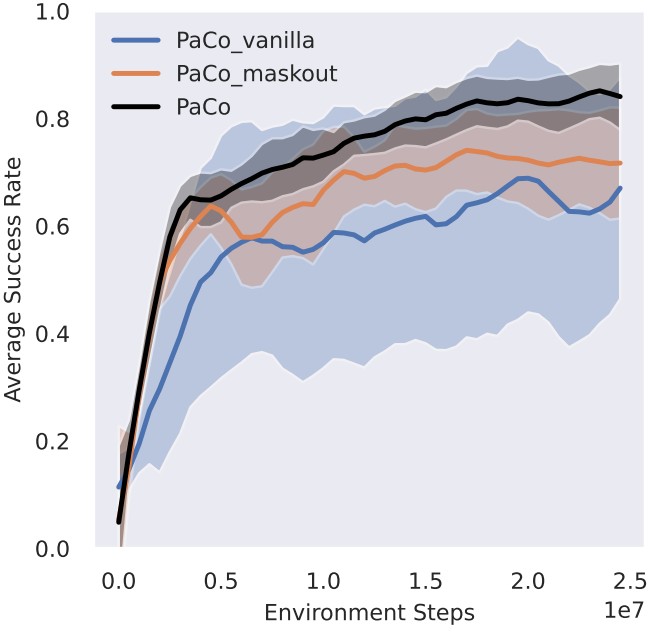

Figure 5: Average success rate curve of PaCo using ablated variations of stabilization schemes.

### A.1.2   Additional Details on Meta-World Benchmarks and Results

For baselines, we used the MTRL codebase [23][9] to produce the results on Meta-World-V2. We tuned the methods on Meta-World V2 [10]. One major change we made is to remove the reward normalizer which was used in [23], leading to better results.

As discussed in the main paper, fixed-goal setting is not practical in real-world usage of robots, it is simpler compared to random-goal setting since we don't need to find the goal-conditioned policy for each skill. Nevertheless, PaCo is able to reach a success rate higher than baseline methods as shown in Table 4.

---

[9] https://github.com/facebookresearch/mtrl   MIT License
[10] https://github.com/rlworkgroup/metaworld/   MIT License

| Methods | Success Rate |
|---|---|
| Multi-Task SAC [36] | $84.5 \pm 12.4$ |
| SAC + FiLM [17] | $74.6 \pm 5.3$ |
| PCGrad [35] | $81.8 \pm 5.2$ |
| CARE [24] | $86.6 \pm 9.8$ |
| **PaCo** | $93.3 \pm 5.8$ |

Table 4: Results on Meta-World-V2 MT10 [36] with fixed goals (MT10-fixed).

MT50 is a more complex benchmark in Meta-World containing 50 different manipulation tasks (including the MT10 tasks). A more complex task combination brings more randomness to the training process and requires more samples and training steps, especially for the random-goal setting (MT50-rand). Therefore it's hard to determine if the policy has reached to the optimal. In Table 5, we show the performance of PaCo with 20 parameter groups and some baselines on MT50-rand in 100M environment steps (for all environments in total).

| Methods | Success Rate |
|---|---|
| Multi-Task SAC [36] | $49.3 \pm 1.5$ |
| SAC + FiLM [17] | $36.5 \pm 12.0$ |
| CARE [24] | $50.8 \pm 1.0$ |
| **PaCo** (K=20) | $57.3 \pm 1.3$ |

Table 5: Results on Meta-World-V2 MT50 [36] with random goals (MT50-rand).

### A.1.3 Additional Results: Performance Scores During Training

In the main paper, we report the final performance on MT10-rand according to the protocol of 20M total environmental for training for 10 tasks (2M environmental steps-per-task). Here we provide the results of different methods at intermediate training steps up to 20M total environmental steps on MT-10-rand for reference. Empirically, we observed that 20M total environmental steps are sufficient for training MTRL methods to convergence and there is no significant improvements with more environmental steps for training.

| Total Env steps | 1M | 2M | 3M | 5M | 10M | 15M | 20M |
|---|---|---|---|---|---|---|---|
| Single-Task SAC | 10.0±8.2 | 17.7±2.1 | 18.7±1.1 | 20.0±2.0 | 48.0±9.5 | 57.7±3.1 | 61.9±3.3 |
| Multi-Task SAC | **34.9±12.9** | 49.3±9.0 | 57.1±9.8 | 60.2±9.6 | 61.6±6.7 | 65.6±10.4 | 62.9±8.0 |
| SAC + FiLM | 32.7±6.5 | 46.9±9.4 | 52.9±6.4 | 57.2±4.2 | 59.7±4.6 | 61.7±5.4 | 58.3±4.3 |
| PCGrad | 32.2±6.8 | 46.6±9.3 | 54.0±8.4 | 60.2±9.7 | 62.6±11.0 | 62.6±10.5 | 61.7±10.9 |
| Soft-Module | 24.2±4.8 | 41.0±2.9 | 47.4±5.3 | 51.4±6.8 | 53.6±4.9 | 56.6±4.8 | 63.0±4.2 |
| CARE | 26.0±9.1 | **52.6±9.3** | 63.8±7.9 | **66.5±8.3** | 69.8±5.1 | 72.2±7.1 | 76.0±6.9 |
| PaCo | 30.5±9.5 | 49.8±8.2 | **65.7±4.5** | 64.7±4.2 | **71.0±5.5** | **81.0±5.9** | **85.4±4.5** |

Table 6: Results on Meta-World-V2 MT10 [36] with random goals (MT10-rand).

### A.1.4 Compositional Vector Visualization

The final output of PaCo framework is a parameter set $\boldsymbol{\Phi}$ of $K$ groups of parameters and the compositional vectors $\mathbf{W} = [\mathbf{w}_1, \mathbf{w}_2, \cdots, \mathbf{w}_T]$ for each skill/task.

In Figure 6 and Figure 7, we show the full compositional matrix used to compose policies for the ten skills in MT10-rand and MT10-fixed experiments. The policies used here reach a success rate of 90% and 100% in the random and fixed goal setting respectively. The absolute value of compositional vectors shouldn't be compared directly since we didn't put any regularization or restriction on the parameter set. We can see not all the columns of the parameter set $\boldsymbol{\Phi}$ play an important rule for each skill, and some skills share very similar compositional vectors.

To better understand the difference of the skills, we also plot the projection of the 5D compositional vector learned in MT10-rand and MT10-fixed in 2D space (referred to as $\mathbf{w}$-space) using Principal

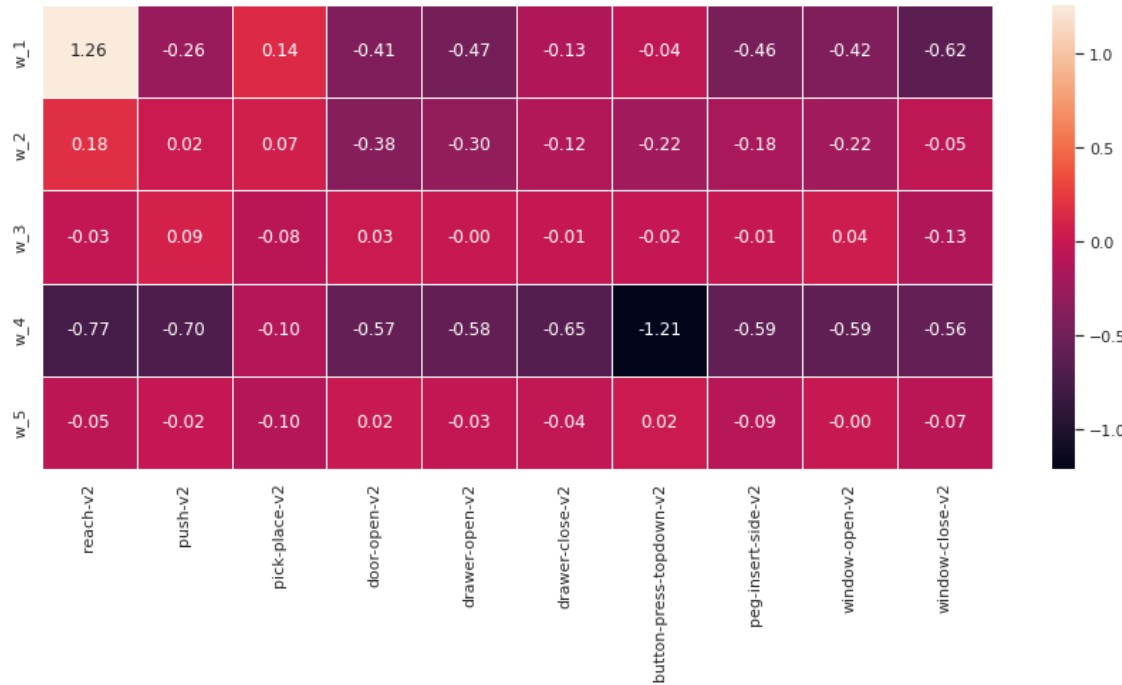

Figure 6: Compositional vector for each task in MT10-rand task. This is a policy reaching average of 90% success rate for all rand-goal tasks (fail on pick-place skill).

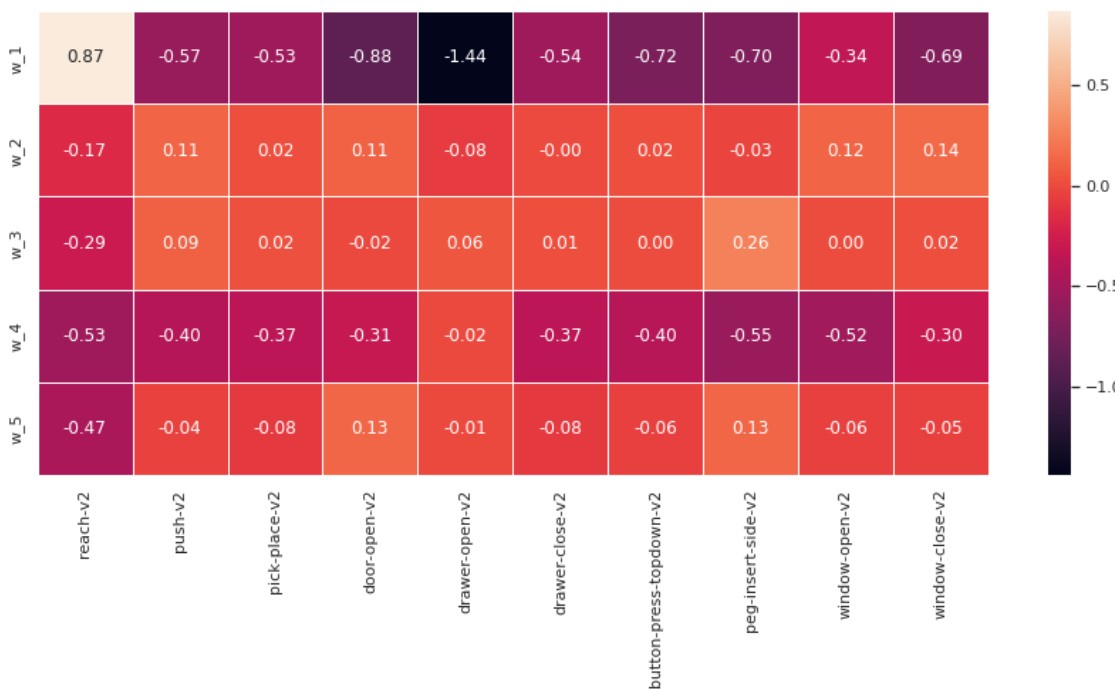

Figure 7: Compositional vector for each task in MT10-fixed task. This is a policy reaching 100% success rate for all fixed-goal tasks.

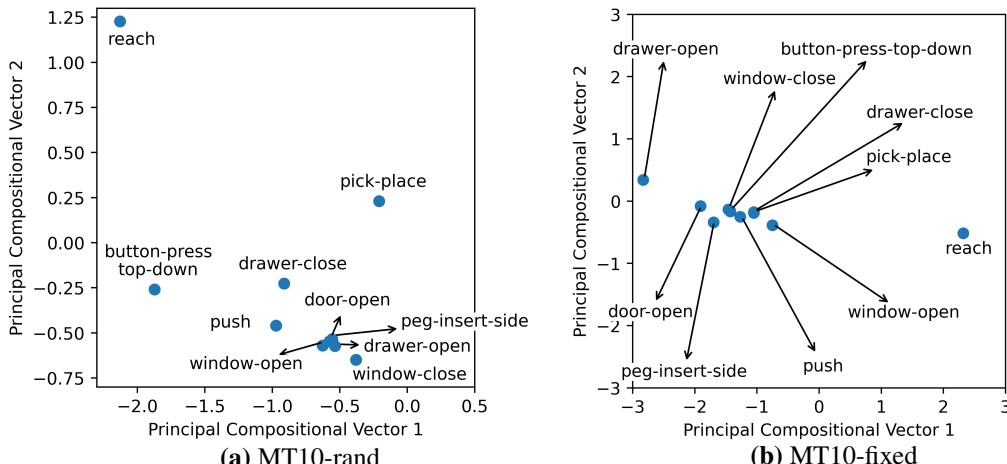

Figure 8: 2D PCA projection of the ten 5D compositional vectors **(a)** learned on MT10-rand (original full vector values are given in Figure 6) and **(b)** learned on MT10-fixed (original full vector values are given in Figure 7).

Component Analysis (PCA). The results are shown in Figure 8. There are several interesting observations from Figure 8:

- Some skills could lie in different part of the **w**-space, *e.g. reach v.s.* others in Figure 8(a);

- Some skills are close in the **w**-space, *e.g. window-open v.s door-open*, *window-open v.s. window-close* in Figure 8(a);

- Another interesting observation is that some skills that are not literally related are also appear to be close in the **w**-space learned by PaCo, *e.g. peg-insert v.s. window-open/window-close/door-open/drawer-open*. Although literally distinct, *peg-insert* and the other skills mentioned above are related from the perspective of behavior, *i.e.*, first interacting with an object (*e.g.*, *peg/window/door*), and then taking a trajectory of motions to accomplish the subsequent operation, (*e.g. insert/window/door*). Therefore, these literally unrelated skills are inherently semantically related at the behavior level. This is something that could be useful but is not able to be leveraged by CARE [24], explaining one possible reason why PaCo is more effective.

- The compositional vectors are more scattered in the MT10-rand case (Figure 8(a)). For the MT10-fixed case (Figure 8(b)), the skills approximately lie on a 1D low-dimensional space, suggesting that there is a possibility to solve all the tasks with a single model, *e.g.* a model with fully shared parameters across tasks. This is likely because of the very limited variations in the MT10-fixed setting and is indeed why we move beyond it with random goals. Indeed, this observation is consistent with our empirical results on MT10-fixed, as shown in Table 4, where single model based methods (*e.g.* Multi-Task SAC) can actually generate very compelling results. Also for a similar reason, the gradient conflicts in the presence of this level of limited task variations are likely to be reconcilable, potentially explaining why gradient-projection-based approach like PCGrad [35] works well for MT10-fixed, but not as good when the level of variations is increased as in MT10-rand.

## A.2 Implementation Details

### A.2.1 Practical Implementation of PaCo

**Compositional Parameters** As introduced in the paper, PaCo keeps the parameter set $\Phi$ and compose the task-specific parameters by $\theta_\tau = \Phi \mathbf{w}_\tau$. In practical implementation, we achieve this compositional structure by replacing the *Linear/Fully-Connect (FC)* layers in neural networks with a new *compositional-FC* layer.

A regular FC layer with input size $d_i$ and output size $d_o$ contains weight $V \in \mathbb{R}^{d_i \times d_o}$ and bias $b \in \mathbb{R}^{d_o}$. Given a batched input $x \in \mathbb{R}^{b \times d_i}$, the output $y$ is calculated by:

$$y = x \cdot V + b \tag{4}$$

In a compositional layer, we add another dimension on the weight and bias, obtaining a parameter set of size $K$, with $\hat{V} \in \mathbb{R}^{K \times d_i \times d_o}, \hat{b} \in \mathbb{R}^{K \times d_o}$. Given a batched input $x \in \mathbb{R}^{b \times d_i}$ and compositional vector $w \in \mathbb{R}^K$, the output $y$ is calculated by:

In this case, the forwarding calculation is

$$y = x \cdot \left( \sum_{i=1}^{K} w_i \cdot \hat{V}_i \right) + \sum_{i=1}^{K} w_i \cdot \hat{b}_i \tag{5}$$

where $\hat{V}_i \triangleq V[i], \hat{b}_i \triangleq b[i]$.

By replacing all the FC layers to compositional-FC layers [11] in the selected networks, we can make the whole structure compositional and flexibly adjust the parameter set size in PaCo. In this way, we don't need to change the implementations and hyper-parameters used in other MTRL methods.

**Random Initialization of Parameters**   For PaCo, we need to initial $K$ groups of parameters with identical structure. Instead of separately initializing all the parameters, we randomly initialize one of the $K$ layers, and copy the weights to the other $K-1$ groups. With the identical initialization on $\mathbf{\Phi}$, all task-specific parameters $\boldsymbol{\theta}_\tau$ will be identical regardless the initialization of $w_\tau$. Experiments show that PaCo can find interpolated policies faster with identical initialization of parameter set.

### A.2.2   MTRL Implementation Details and Hyper-parameters

In this section, we provide the hyper-parameter PaCo used in MT10-rand experiment in Table 7, and some general hyper-parameters used across PaCo and the baselines in Table 8.

| Hyper-parameter | Value |
|---|---|
| extreme loss threshold $\epsilon$ | 3e3 |
| param-set size $K$ | 5 |
| compositional vector learning rate | 3e-4 |

Table 7: PaCo specific hyper-parameters on MT10-rand

| Hyper-parameter | Value |
|---|---|
| batch size | 1280 |
| number of parallel env | 10 |
| MLP hidden layer size | [400, 400, 400] |
| policy learning rate | 3e-4 |
| Q learning rate | 3e-4 |
| discount | 0.99 |
| episode length | 150 |
| exploration steps | 1500 |
| replay buffer size | 1e6 |

Table 8: General MTRL hyper-parameters on MT10-rand

### A.2.3   Details on Computational Resources

For training, we used internal cluster with GeForce RTX 2080 Ti GPU. Training is repeated 10 times with different seeds. On MT10-rand for baseline methods, the time required for a single complete run varies from 10 (*e.g.* Mult-Head SAC [36]) to 31 hours (*e.g.* PCGrad [35]). For PaCo, the time takes for each run ranges from 20 to 30+ hours, depends on the compositional structure used. One point to note is that for the baseline methods, there is a non-negligible possibility of failure in training, when the training loss explodes. In this case, a new training job has to be relaunched thus consuming additional computational resources. For more details on training stability, please refer to Section A.2.4.

---

[11]Although only MLP is used in this work, compositional conv layers can be implemented in a similar way.

### A.2.4 Loss Explosion and Training Stability

It has been observed empirically that the MTRL may suffer from instability in training, sometimes with exploding loss. This is aligned with the know issue of gradient conflictions between different tasks [35]. To mitigate this issue, [24] adopted an empirical trick to stop training once this issue is spotted.[12] This whole run will be discarded and and new training run need to be launched instead. This scheme has been applied to *all the baseline methods* in this work, following the setting in [24]. Note that whenever this happens, although not taken into account by following the *stop-relaunch* trick from [24] when reporting the performance, the actual effective number of environmental steps is increased.

This scheme is unnecessary for PaCo, since it has an built-in scheme for stabilization, leveraging its unique decompositional structure.

### A.3 Initial Attempts on Some Possible Future Extensions with PaCo

### A.3.1 PaCo-based Transfer Learning

Going beyond MTRL, another question we may ask in application is how we benefit from the policy trained by PaCo when we meet new tasks. The unique property of a well separated task-agnostic parameter set and task-specific compositional vector give us potential to use PaCo in a more challenging continual setting. The main reason for catastrophic forgetting in continual learning is that the training on new tasks modifies the policies of existing tasks. However, in our PaCo framework, if we can find the policies for new task $\tilde{\tau}$ in the existing policy subspace defined by $\mathbf{\Phi}$ with a new compositional vector $\mathbf{w}_{\tilde{\tau}}$, the forgetting problem can be avoided. With no change on $\mathbf{\Phi}$, we extend the existing parameters to a new task with no additional cost. In reality, there is no guarantee for the existence of such policy, the relation between skills are quite important. However, in experiments, we do find successful extensions from existing skill set to a new skill when the skills are similar. For instance, *reach, door-open, drawer-open* **to** *drawer-close*.

In practice, we can design a more general training scheme to learn the policy for a series of tasks. Given a parameter set $\mathbf{\Phi}$ with $K$ parameter groups trained on $N$ tasks, if we find the policy for new tasks in the policy subspace, we save the compositional vector for the new task. If we cannot find the policy in subspace, we train the new tasks on a new parameter set $\tilde{\mathbf{\Phi}}$ and merge them into subspace with higher dimension. Verifying this property on larger skill sets is an interesting future direction and requires more complex experiment designs.

---

[12] https://github.com/facebookresearch/mtrl/blob/eea3c99cc116e0fadc41815d0e7823349fcc0bf4/mtrl/agent/sac.py#L322