# OpenReview forum: "PaCo: Parameter-Compositional Multi-task Reinforcement Learning"
_NeurIPS.cc/2022/Conference — NeurIPS 2022 Accept_

### Official Review · Reviewer_U1FU · 2022-07-05

**Rating:** 6
**Confidence:** 3
**Soundness:** 2 fair
**Presentation:** 3 good
**Contribution:** 3 good

**Summary:**

Authors propose a simple, yet effective way to learn parameters in multi-task RL setting, where parameters for a task are a task-specific linear combinations of task-agnostic parameters. Parameters of linear combinations as well as task-agnostic parameters are both learned end to end. This simplicity allows for resets of linear combination parameters for tasks with exploding losses, which should lead to more stable learning. To this end, authors propose to use a random convex-combination of non-exploding linear combination parameters as a stabilization scheme.

**Questions:**

- In lines 65-66 you mention that you use separate temperature for each task with different skills. Does that mean that you use 10 different values of alpha in MT10-rand?
  - If yes: Why is this necessary? Is the performance sensitive to the choice of these parameters and would it degrade with single value?
  - If no: Can you please explain what you meant by this?
- I am wondering about the relation to soft-modularization. As far as I can tell it seems to be more general because it allows for dependency on s. Your method should then be a specific case of this ($z$ does not depend on s and are shared for all $m$). Why should it then be less flexible (line 189.)?
- I am wondering about the performance of single-task model where you use separate network for each task (line 175). Have you tried to run such baseline?
- How does the value of parameter epsilon affect performance? Have you tried runs with different values? If yes what was the performance?
- I am surprised that in table 3 K=8 performs worse than K=5 since the model with K=8 should be able to represent anything that K=5 can. In the footnote you also mention that K=10 performs better than K=5. Do you have performance metrics for other K? Why does high K lead to worse results(you mention additional complexities from over-parametrization, could you elaborate on that)?
- Are the networks used for other baselines the same? Do all methods have same amount of hyper-parameters?


**Limitations:**

I do not foresee any negative social impact. Limitations of the work seem to be addressed as far as I can tell.

**Strengths And Weaknesses:**

I think that the paper is well written and presents a reasonably novel and interesting idea with surprising results. I do have some questions about experiments and baselines and I am a bit surprised that the algorithm performs worse with more parameters. Therefore I am not certain if more experimental evaluation is not needed. I also think that the contribution and the paper could be strengthened by some theoretical results/motivation or intuition about why the particular proposed reset scheme increases the performance by so much (10-15%). I do not yet have a very strong opinion about whether the paper should be accepted although I am leaning more towards acceptance. I am willing to change my score based on the discussion with authors and fellow reviewers.

**Strengths**:
- Paper is in general well written and clear, the setting and method are well explained. I liked section 3 in particular
- In my view the idea seems novel enough even though part of it is closely related to previous approaches (specific case of Soft-Modularization?). However, I am not very familiar with current Multi-task RL approaches so I am not 100% certain about this.
- The results indicate that proposed re-initialization is quite effective on a common Meta-world benchmark (MT10)

**Weaknesses**:
- Only one domain is considered which might make it hard to judge the significance and scalability of the results especially since one of the ablations is a bit counter-intuitive (see questions). Appendix seems to contain experiments from a larger MT50 domain. However, these are not part of the main text.
- Experiment are mostly sound and include ablations. However, I do have several questions about them and would like to (among others) see learning curves from the experiments or intermediate/max values of the performance metrics instead of just final values

**Comments/Suggestions/Typos (do not affect the score):**
- line 80: use former one instead of previous one?
- line 84: I am not sure if about "share similar dynamics (actions space)" in this case since I presume action space is shared in general. Maybe transition function could fit better in this context?
- line 156 $\eta$ is used again but denotes something different than in background section as far as I can tell
- line 159: "compositional natural" nature?
- line 331 "tasks covers" cover

_______________________________________________________________
Update: I believe that the concerns about the domain and missing intermediate scores were sufficiently addressed by the authors. I have therefore increased my score. However, after observing the results, I also agree with reviewer CCrp that seeing the results beyond 2 million would be helpful. This is because while some method (SAC+ Film, PCGrad) have already converged, others (Care, Paco, single-task SAC) seem to still be improving so it would be nice to see how training progresses further and if/how long Paco keeps an advantage.

---

> ### Author Response · Authors · 2022-08-02
> **Response to Reviewer U1FU (Part 3/3)**
>
> **Q5: I am surprised that in table 3 K=8 performs worse than K=5 since the model with K=8 should be able to represent anything that K=5 can. In the footnote, you also mention that K=10 performs better than K=5. Do you have performance metrics for other K? Why does high K lead to worse results (you mention additional complexities from over-parametrization, could you elaborate on that)?**
>
>
>
> To interpret the results in Table 3a properly, it is very important to understand the following two aspects:
>
> - **The role of K and its impacts on performance.** K can be intuitively understood as a hyper-parameter for adjusting the strength of parameter sharing. Different values have different impacts on parameter-sharing and sample efficiency.
>
>     - Smaller K (e.g., much smaller than task number T) will have stronger enforcement on parameter sharing. It can be observed that a too strong parameter sharing setting (e.g., K=3 in Table 3a) will limit its performance due to its over-constrained policy parameter space representable by $\mathbf{\Phi}\mathbf{w}$.
>
>     - Larger K (e.g., comparable to task number T) offers a larger representation space (more parameters) but at the same time enforces less on parameter sharing, which will decrease sample efficiency (i.e., learns slower).
>
> - **The MTRL evaluation setup.** For the standard MTRL setup, we compare methods at a given fixed number of environment steps for training (e.g., 2M/task established in previous work where most methods reach their convergence performance).
>
>
> Once these two points are clear, the results in Table 3a would not be “surprising” but “expected”, since both too small or too large K will decrease the performance (at 2M), although due to different reasons. More concretely, a too small K decreases the performance due to the over-constrained policy parameter space. On the other hand, a too large K will reduce sample efficiency since it encourages less on parameter-sharing. Therefore, under the standard MTRL setting, where we evaluate the models after being trained on a fixed number of environmental steps, the model will typically have lower performance because of its lower sample efficiency.
>
>
> The results in the footnote on K=10 with special one-hot initialization for w-vectors is another empirical result that supports the analysis above. It implies that the lower performance for large K is due to reduced sample efficiency and not due to limited capacity (as in the small K case) since there exist solutions with better performance for large K case (but might need some additional requirements to achieve, e.g., better initialization, additional regularization, etc.).
>
>
>
>
> **Q6: Are the networks used for other baselines the same? Do all methods have same amount of hyper-parameters?**
>
> The network structures and hyper-parameters for the common components (e.g., SAC algorithm related, network structure related) are the same across all the methods (as summarized in Appendix Table 4). For soft-modularization and CARE which introduce extra networks and network structures, we use the hyperparameters reported in their paper.
>
> Regarding the amount of hyper-parameters used, most hyper-parameters are shared across methods. Specific parameters for CARE are the number of task encoders and encoder size, and specific parameters for soft-modularization are the structure sizes of the routing network. PaCo has two specific hyperparameters (listed in Appendix Table 3): extreme loss threshold epsilon (as discussed in response to your question **Q4**) and parameter set size K (as discussed in response to your question **Q5**).
>
>
>
>
>
> **Comments on Typos.**
>
> It is indeed more appropriate to use “transition function” in L84. Thanks for the suggestion. We will fix it.
>
> Thanks for pointing out the notation issue of $\eta$ as well as other typos. We will fix them in revision.
>
>
>
>
>
> **References**
>
> [CARE] Multi-Task Reinforcement Learning with Context-based Representations, ICML 2021
>
> [SoftModule] Multi-Task Reinforcement Learning with Soft Modularization, NeurIPS 2020

---

> ### Author Response · Authors · 2022-08-02
> **Response to Reviewer U1FU (Part 2/3)**
>
> **Q1: In lines 65-66 you mention that you use separate temperature for each task with different skills. Does that mean that you use 10 different values of alpha in MT10-rand? Why is this necessary? Is the performance sensitive to the choice of these parameters and would it degrade with single value?**
>
>
> Your understanding is correct. Different temperatures are used (and auto-adjusted as in SAC) for different tasks and the performance will degrade with a single value. This is actually a standard setup established in previous MTRL methods (e.g., SoftModule) and also adopted in later work such as CARE (appendix of CARE paper Table 9-15, temperature is “learned and disentangled with tasks” for all methods). It is used because different tasks may have different learning dynamics along the training process. In this work, we simply follow the setting established in previous work and also apply it to all methods. We apologize that we did not make it clear in the paper. We will revise the paper and clarify this point.
>
>
>
> **Q2: I am wondering about the relation to soft-modularization. As far as I can tell it seems to be more general because it allows for dependency on s. Your method should then be a specific case of this (z does not depend on s and are shared for all m). Why should it then be less flexible (line 189.)?**
>
> Soft-modularization divides each layer into several groups of “modules” and then combines their outputs with “soft weights” from another “routing” network. To obtain these soft weights, the routing network takes both the task id and state as input.
>
> In PaCo, the w-network generates a compositional vector $\mathbf{w}$ by taking only the task id as input, and  the compositional vector  is used for combination in the parameter space ($\boldsymbol{\theta} =\mathbf{\Phi}\mathbf{w}$).
>
> While “dependency on state” is one difference between PaCo and soft-modularization, another crucial difference is that PaCo interpolates in parameter space, while soft-modularization performs combination in the network output space (mentioned in Line186-187). Because of this, even if we incorporate s into the w-network of PaCo, it is still different from soft-modularization. As a side note, it’s easy to incorporate s into w-network, but doing so will lose the separation property of task-specific and agnostic parts, which contributes to the final performance of the proposed method (w-reset).
>
> The Equation in L185 of the paper was meant to help capture the connections between two methods and has to be interpreted together with the sentences in L186-187. But unfortunately, it seems that this way of presentation could be misleading if looking at the Equation itself. Because of this, we will remove the equation in revision and use literal descriptions for the connection and differences to avoid confusion.
>
> We agree the term “less flexible” might be a bit misleading. By “less flexible in some cases”, we actually meant “less desirable/applicable in some cases”. For example, in cases where we want to conduct some task-specific operations, e.g., w-reset, it is not straightforward to do so in soft-modularization since we cannot find parameters that are dedicated to a specific task, because of the fact that all the networks have a “dependency on state”. We will clarify this point in revision.
>
>
>
>
>
> **Q3: I am wondering about the performance of single-task model where you use separate network for each task (line 175). Have you tried to run such baseline?**
>
> Thanks for the suggestion. We have now added this Single-Task SAC baseline according to your comment. Single-Task SAC achieves 61.9% average success rate on MT10-rand under the standard setting (2M environment step/task for training). Please refer to the reply to “**Common question on Single-Task SAC**” above for more details.
>
>
>
> **Q4: How does the value of parameter epsilon affect performance?**
>
> Epsilon is the threshold for detecting abnormal losses. Since extreme loss values are very large (>1e3 and could explode to even >1e7) compared to normal values (<1e3), its value is relatively easy to set and the performance is not sensitive to different values within a reasonable range.
> In practice, we’ve tried setting it to {3e3, 5e3} without observing significant differences in performance, and both provide similar improvements over PaCo without w-reset.

---

> ### Author Response · Authors · 2022-08-02
> **Response to Reviewer U1FU (Part 1/3)**
>
> We thank the reviewer for the positive review and insightful comments.
>
> Responses to your questions are below.
>
>
>
> **Weakness 1: Only one domain is considered … Appendix seems to contain experiments from a larger MT50 domain. However, these are not part of the main text.**
>
> MetaWorld is a well-established benchmark with open source reproducible baselines developed in the MTRL community, with various types of robotic manipulation tasks. It has been widely used in previous MTRL research work (e.g. SoftModule, CARE) as the environment for experimental evaluations.
> Apart from the already included various of manipulation skills in the original MeteWold setting, we further expanded it with goal variations, and actually have experimented on 3 different settings in total: apart from MT-10-rand (10-tasks with random goals) reported in the main paper, we do have results on MT-10-fixed (10-tasks with fixed goals) and MT-50-rand (50 tasks with random goals) in the appendix. We will move some of these results to the main paper.
>
>
>
> **Weakness 2: Intermediate values of the performance metric.**
>
> The reason we reported the final performance at 2M environmental frames/task for training is that it is a standard protocol established in previous MTRL work [SoftModule, CARE] and we followed this protocol in the paper. Beyond this standard protocol, according to your request, we have now provided the results with intermediate values on MT-10-rand in the table below
>
>
>
> | Env steps (per task) | 100K | 200K | 300K | 500K| 1M | 1.5M | 2M |
> |:-----------------------:|:-------:|---:|:---:|:---:|:---:|:---:|:---:|
> | Single-Task SAC | 10.0±8.2 |17.7±2.1 |18.7±1.1|20.0±2.0 | 48.0±9.5 | 57.7±3.1 | 61.9±3.3 |
> | Multi-Task SAC |**34.9±12.9**|49.3±9.0|57.1±9.8|60.2±9.6|61.6±6.7|65.6±10.4|62.9±8.0|
> |SAC + FiLM|32.7±6.5|46.9±9.4|52.9±6.4|57.2±4.2|59.7±4.6|61.7±5.4|58.3±4.3|
> | PCGrad | 32.2±6.8 |46.6±9.3|54.0±8.4|60.2±9.7|62.6±11.0|62.6±10.5|61.7±10.9|
> |Soft-Module|24.2±4.8|41.0±2.9|47.4±5.3|51.4±6.8|53.6±4.9|56.6±4.8|63.0±4.2|
> |CARE|26.0±9.1|**52.6±9.3**|63.8±7.9|**66.5±8.3**|69.8±5.1|72.2±7.1|76.0±6.9|
> |PaCo|30.5±9.5|49.8±8.2|**65.7±4.5**|64.7±4.2|**71.0±5.5**|**81.0±5.9**|**85.4±4.5**|
>
>
>
>
> **Suggestion: the paper could be strengthened by some intuitions on why w-reset is effective.**
>
> Thanks for the suggestion. Since MTRL methods share parameters across different tasks, there are cases where the extreme loss/gradient from one task severely affect the whole training (the low performance and high variance of PaCo-vanilla version), some previous works/code implementations early stop the training when such situation occur (mentioned in L152-154).
>
> In our work, beyond purely designing the network structure for parameter sharing, we also try to stabilize the training using the benefits we get from the PaCo framework. First, ignoring the loss (loss mask-out) excludes the contribution of the particular exploding loss on parameter updates and the potential impact of this extreme task loss on other tasks, but this could also compromise the overall performance because of the stopped learning on the masked-out tasks (as mentioned in Line 293-295) since there is no direct gradient for fixing that particular task and it will likely to be always excluded from training from then on.
>
>
>
> On the other hand, w-reset offers an opportunity to continue learning for the task with exploding loss, by resetting its w-parameter, which will generate a new task parameter vector
> when composed with $\mathbf{\Phi}$  as
> $\mathbf{\Phi} \mathbf{w}_{\rm new}$.
>
> At the same time, this reset has no influence on other non-exploding tasks since $\mathbf{\Phi}$ and their corresponding $\mathbf{w}$ (and therefore the composed parameters) remain the same. This offers a new starting point to further learn on the task with previously exploding loss, which naturally provides opportunities to further improves over loss-maskout, as shown in Table 2. This is a simple scheme but we are able to use it due to the separation of task-specific and task-agonistic parameters in the PaCo method.
> We will add these analysis into the revised paper to facilitate understanding.

---

### Official Review · Reviewer_DW33 · 2022-07-08

**Rating:** 6
**Confidence:** 4
**Soundness:** 2 fair
**Presentation:** 4 excellent
**Contribution:** 3 good

**Summary:**

The paper proposes PaCo, a framework for multi-task learning in RL. It uses a set of parameters that are composed in a unique way for each task. A compositional vector w is learned for each task, which weights the items in the parameter set. This results in a weighted policy for each task that relies on shared underlying parameters. This can help with learning shared structure across tasks and improve sample efficiency.

**Questions:**

* Are all experiments using 10 seeds? I see that table 1 is using 10 seeds, but I'm not sure about tables 2 and 3. For example, the std of PaCo-vanilla in table 2 seems very high and I'd expect it to be lower with 10 seeds.

* Page 7, footnote 3: If these baselines were implemented for a different setting, are there any hyperparameters that need to be re-tuned for MetaWorldV2? It would be helpful to know which hyperparameters from appendix table 4 were tuned for PaCo specifically and what values for each were included in the search.

* I'd like to see a baseline where each task has its own policy (no parameter sharing) and another baseline where the size of the parameter set (K) is the same as the number of tasks. In the K=10 case for MT10, for example, does each task essentially claim its own item in the parameter set and there is limited parameter sharing?

* I'm curious about the importance of assigning/tuning separate temperatures for each task (L89). Is this a crucial aspect of PaCo's success over baselines?

Minor (no need to comment on the below):
* Appendix section 3.1: I don't think this section is that important, but if it's going to stay in the paper, please add a figure with evidence of this experiment (e.g. reward curves for PaCo and the modified ratio PaCo).

* L313: "Overall, the increasing in parameter set size K grows with the task number n but essentially at a lower speed." I don't see enough evidence for this claim given that the parameter set size ablation is only shown for MT10 (10 tasks). I would probably remove the claim unless the parameter set size ablation can be repeated on MT20 or MT50 (to show how it scales).

Typos:
* L136: details about PaCo is --> details about PaCo are
* L159: compositional natural --> compositional nature
* L294: some task already explodes --> some task has already exploded

**Limitations:**

I appreciate the limitation discussion of the linear compositional form.

**Strengths And Weaknesses:**

The work is original and clear with strong results on a multitask benchmark. If my concerns are addressed (e.g. number of seeds, hyperparameters, and a couple of unsupported claims), I support the acceptance of this paper.

Strengths
* Parameter sharing is an interesting area for improving sample efficiency in MTRL, and I have not seen published work that composes policies in this way at the parameter level.
* The interpolated policy visualization in Figure 3 is unique and interesting.

Weaknesses
* One of the potential benefits of this framework is the ability to add new tasks after training the parameter set, which is mentioned on L144 and described in appendix section 3.2 (which says "in experiments, we do find successful extensions from existing skill set to a new skill when the skills are similar"). This claim is unfortunately unsupported. Please either provide evidence of such experiments or do not include the claim.

-------
Update: I thank the authors for their responses. The concerns (especially about number of seeds and hyperparameters) were sufficiently addressed; thus I am increasing my score from borderline accept to weak accept. I do recommend expanding the discussion/analysis of new results, i.e. "continual learning without forgetting" (what to compare these numbers to) and single-task SAC vs K=10 vs K=10 with one-hot initialization (what relative performance numbers mean for these three versions).

---

> ### Author Response · Authors · 2022-08-02
> **Response to Reviewer DW33 (Part 2/2)**
>
> **Q3: I'd like to see a baseline where each task has its own policy (no parameter sharing) and another baseline where the size of the parameter set (K) is the same as the number of tasks. In the K=10 case for MT10, for example, does each task essentially claim its own item in the parameter set and there is limited parameter sharing?**
>
>
>
>  Thanks for your suggestion and we have now added this baseline (Single-Task SAC). Single-Task SAC achieves 61.9% average success rate on MT10-rand under the standard setting (2M environment step/task). For more details on Single-task SAC, please refer to the reply to “**Common question on Single-Task SAC**" above.
>
>
> K can be intuitively understood as a hyper-parameter for adjusting the strength of parameter sharing. Different values have different impacts on parameter-sharing and sample efficiency.
> - Smaller K (much smaller than task number T) will have a stronger enforcement on parameter sharing. It can be observed that with a too strong parameter sharing setting (e.g. K=3 in Table 3a) will limit its performance due to its over-constrained policy parameter space representable by $\mathbf{\Phi} \mathbf{w}$.
>
> - Larger K (e.g. comparable to task number T) offers a larger representation space, but at the same time enforces less on parameter sharing, which will decrease sample efficiency (i.e., learns slower).
> For the standard MTRL setup which compares methods at a given fixed number of environment steps for training (e.g. 2M/task established in previous work), both too small or too large K will decrease the performance: i) a too small K decreases the performance due to the over-constrained policy parameter space; ii) a too large K will reduce sample efficiency since it encourages less on parameter-sharing. In the standard MTRL setting where we evaluate the models after being trained on a fixed number of environmental steps, the model will typically have a lower performance because of its lower sample efficiency.
>
> Therefore, in practice, K with a medium value intuitively should be better for performance as it can strike a better balance between parameter-sharing and representation power, which impacts sample efficiency and performance. This is aligned with the results in Table 3a of the paper.
>
>
> Because of these reasons, in the extreme case when K equals the task number (K=10 for MT-10), the performance will be worse than that of a medium K after being trained on 2M/task, due to decrease in sample efficiency. Since the w-vectors are randomly initialized, it is very unlikely for them to converge to the “one-hot solution” (where each task “claims its own item in parameter set”) without additional constraints on w.
>
> This actually inspires us to further think on how to incorporate more constraints on w for future work. In the paper (page 8 footnote 5), we actually have related results for K=10 under one possible form where we initialize w-vectors to be one-hot and obtain improved performance (88.9 ± 1.9). More advanced forms of regularization might need to incorporate inter-task similarities for guidance. A more in depth investigation of this topic is out of the scope of the current paper but is very interesting and we will conduct it in future work.
>
>
>
> **Q4: I'm curious about the importance of assigning/tuning separate temperatures for each task (L89). Is this a crucial aspect of PaCo's success over baselines?**
>
> We apologize that we did not make it clear in the paper. Assigning and (auto)-tuning of separate temperatures for each task is a standard setup used in previous MTRL methods (e.g. SoftModule, CARE). For example, in CARE work, this is applied to all the baselines as well (appendix of CARE paper Table 9-15, temperature is “learned and disentangled with tasks” for all methods). It is typically used because different tasks may have different learning dynamics along the training process.
> We follow this setting in CARE to apply it to *all methods*. We will make this clear in the revised paper.
>
>
>
>
>
> **Minor comments.**
>
> Thanks for the comments and we will modify the paper as you suggested. We will remove Appendix section 3.1 since it is less relevant to the main theme of the paper and also L313 as you suggested. And thanks for pointing out some typos, we will fix them in the revised paper.
>
>
>
>
>
> **References**
>
>
> [CARE] Multi-Task Reinforcement Learning with Context-based Representations, ICML 2021
>
> [SoftModule] Multi-Task Reinforcement Learning with Soft Modularization, NeurIPS 2020

---

> ### Author Response · Authors · 2022-08-02
> **Response to Reviewer DW33 (Part 1/2)**
>
> We thank the reviewer for the positive review and appreciation of our work.
>
> Responses to your questions are below:
>
>
>
>
> **Weakness: unsupported claim “in experiments, we do find successful extensions from existing skill set to a new skill when the skills are similar” in appendix section 3.2.**
>
>
>
> According to your comment, we now provide below some preliminary but encouraging results on the effectiveness of PaCo for continual learning without forgetting, by fixing already learnt $\mathbf{\Phi}$ and learning only a new $\mathbf{w}$ vector for the continual task. For a parameter set (K=2) trained on a set of Trained-Tasks, we can obtain policy for a new task by reusing the same $\mathbf{\Phi}$ and only learning a new $\mathbf{w}$ vector, with reasonable performance, as shown in the table below.
>
>
>
> The results are encouraging from the perspective that it demonstrates the possibility of reusing a previously learned $\mathbf{\Phi}$ for solving a relevant new task by optimizing for a low dimensional $\mathbf{w}$ vector (here $\mathbf{w}$ is a 2d vector) only.
>
>
>
>
> | Trained Tasks | New Task | Success Rate on New Task (%) |
> |:---------------------------------|:--------------------:|:------:|
> | {reach, door-open, drawer-open} | drawer-close | 75 ± 9 |
> | {window-open, window-close, door-open} | door-close | 90 ± 5 |
>
>
>
>
>
>
> In another different but related setting, when desirable, we can also allow $\mathbf{\Phi}$ for fine-tuning. For example, for {reach, door-open, drawer-open} → drawer-close, we can further boost the success rate from 75% to close to 100% with fewer steps compared to single task training, when using the previously learned $\mathbf{\Phi}$ for initialization and fine-tuning from there.
>
>
>
> These results imply that the previously learned parameter set $\mathbf{\Phi}$ could be useful in learning some new skills depending on the tasks used in the training set.
>
>
>
> While encouraging, it is important to note that these are some initial, non-extensive results on some possible future extensions, and is not an essential part of the main theme of this paper, which is on the parameter-compositional MTRL method itself. Therefore, we will include them in the appendix in revision to avoid the distraction it might bring to the main theme of the paper.
>
>
>
>
>
> **Q1: Are all experiments using 10 seeds? I see that table 1 is using 10 seeds, but I'm not sure about tables 2 and 3. For example, the std of PaCo-vanilla in table 2 seems very high and I'd expect it to be lower with 10 seeds.**
>
>
>
> Yes. The high std of PaCo-vanilla in Table 2 is actually caused by the unstable nature of PaCo-vanilla itself, since there are no stabilization schemes used (no stop-relaunch, no loss-maskout, no w-reset, as mentioned in paper L293-294). In fact, two of the ten seeds have a success rate lower than 0.3 (others greater or around 0.7), which results in the large std, showing the unstable nature of MTRL training as commonly observed in MTRL, which implies why stabilization schemes are necessary for MTRL in practice.
>
>
>
>
> **Q2: Page 7, footnote 3: If these baselines were implemented for a different setting, are there any hyperparameters that need to be re-tuned for MetaWorldV2? It would be helpful to know which hyperparameters from appendix table 4 were tuned for PaCo specifically and what values for each were included in the search.**
>
>
>
> The hyper-parameter values in Table 4 are inherited from the standard setup used in previous MTRL work (e.g. Table 9 in appendix of CARE paper) and are not specifically tuned for PaCo. The same hyper-parameter values are used across all the methods.
>
>
>
> The major difference between Meta-World-V2 and V1 is that V2 updated reward design with more balanced scales across tasks. On V1, to balance reward scales, a reward normalization wrapper is typically used in previous methods. Moving to V2, since reward balancing is built into the environments, the reward normalization is unnecessary, and the rest of the algorithmic part can be left intact. We also tested different hyper-parameter values but did not observe substantial differences in performances. We therefore keep them to be consistent with previous work.
> We will add these to implementation details of the revised paper.
> Furthermore, to facilitate fellow researchers on future research, we will also release the code for reproducing our results with an updated url in the paper.

---

> ### Author Response · Authors · 2022-08-08
> **Thanks for your positive feedback and increasing the score**
>
> It's great to know that we addressed your concerns and thank you for increasing the score of the paper.

---

### Official Review · Reviewer_CCrp · 2022-07-12

**Rating:** 5
**Confidence:** 4
**Soundness:** 2 fair
**Presentation:** 3 good
**Contribution:** 2 fair

**Summary:**

The paper proposes an approach for composing parameters for solving multi-task RL problems and obtains promising results on MetaWorld benchmark

**Questions:**

1. The paper mentions a "stop-relaunch" trick ion [23] but I could not find a reference to that trick in [23[.

2. In line 154, it says " in PaCo, there is a natural way to mitigate this issue without resorting to ad-hoc [23] or more expensive schemes [34]." Isnt the loss maskout another adhoc trick. Infact this trick also introduces a new hyper-parameter.

3. I am not sure I understand why w-reset is used. I understand that we can do it but isnt PaCO already ignoring the loss when it is greater than $\eps$. I didnt understand the need for both loss masking and w-reset

4. It would be helpful if the authors can comment on the "novelty" of the work. Is the main novelty that they are doing composition in the parameter space, instead of the function space? I am not considering this as a weakness - I just want to make sure I am not missing anything here.

5. Could the authors add details about the computation overhead of the different approaches?

6. The analysis related to Figure 4 is not very convincing. eg arguing that "peg-insert v.s. window-open/window-close/door open/drawer-open are related from the perspective of behavior, i.e., first interacting with an object (e.g., peg/window/door), and then taking a trajectory of motions to accomplish the subsequent operation, (e.g. insert/window/door)" does not sound convincing as the same argument applied to "pick-place" while it is quite far in the latent space.

**Limitations:**

Covered in section 7

**Strengths And Weaknesses:**

## Strengths

1. It is a relatively straightforward approach that works well in practice.
2. The paper is mostly well written (though see next section)
3. The baselines are both recent and relevant.

## Weakness

1. The paper should add the "one-model-per-task" baseline to their results, to get a sense of performance gap.

---

> ### Author Response · Authors · 2022-08-02
> **Response to Reviewer CCrp  (Part 2/2)**
>
>
>
>
>  **Q5: Computational overhead of different methods.**
>
>
> For a total of T tasks, PCGrad has a computational complexity of O(T^2), as it needs to compute pairwise task gradient inner products for resolving gradient conflicts. The rest methods have a computational complexity of O(T).
>
> The number of network parameters also contributes to the actual computational cost. Methods using simple backbones (e.g., Multi-Head SAC, Multi-Task SAC, FiLM) are usually faster. Other methods have some additional parameters because of their introduced additional modules on top of the same backbone. For example, CARE’s total number of network parameters is affected by the number of encoders in the mixture of encoder networks for processing inputs; SoftModule is affected by the size and number of modules in each layer. PaCo’s total number of network parameters is affected by the size of the parameter set (K).
>
> **Q6: The analysis related to Figure 4 is not very convincing … as the same argument applied to "pick-place" while it is quite far in the latent space.**
>
>
>
> This is a very good point that we missed mentioning in the paper, and thanks for pointing it out. Actually, it also inspires new ideas for future research.
>
>
>
> Firstly, Figure 4 is the 2D PCA projection of the 10 compositional parameters $\mathbf{w}$ for {reach, push, pick-place … } obtained by training PaCo on MT-10-rand, with ~85.4% success rate (Table 1). Actually, among all the 10 tasks, pick-place is a task that has a low success rate after training. Because of this, the $\mathbf{w}_{\rm pick\-place}$ is not a representative $\mathbf{w}$ for solving the pick-place task. We conjecture that this is the main reason why it is not close enough to other points with similar behavior patterns in terms of motion trajectories.
>
>
>
> Secondly, your question actually inspires us to think further about how to incorporate prior task similarity information (when available) into learning in a general way. In this case, for example, if we have the prior knowledge that "pick-place" is more similar to tasks such as window-open/window-close/door-open/drawer-open than some other task (e.g., button-press top-down), we may have an opportunity to guide the model to learn a better policy for "pick-place". Of course, this is our current intuition only as the inter-skill similarities are typically unavailable in standard MTRL settings. Nevertheless, how to leverage the task similarities when provided or even learn it together is a very interesting direction, and we will explore it in future work.
>
>
>
> **References**
>
> [CARE] Multi-Task Reinforcement Learning with Context-based Representations, ICML 2021
>
> [FiLM] FiLM: Visual reasoning with a general conditioning layer, AAAI 2018
>
> [PCGrad] Gradient Surgery for Multi-Task Learning, NeurIPS 2020
>
> [SoftModule] Multi-Task Reinforcement Learning with Soft Modularization, NeurIPS 2020

---

> ### Author Response · Authors · 2022-08-02
> **Response to Reviewer CCrp  (Part 1/2)**
>
> We thank the reviewer for the efforts in reviewing our work and the comments that helped to further improve the quality of the paper. Responses to the questions are below:
>
>
>
>
>
> **Weakness: The paper should add the “one-model per task baseline” to their results.**
>
>
>
> Thanks for the suggestion. We have now added the one-model per task baseline (Single-Task SAC) as you suggested. Single Task SAC reaches an average success rate of 61.9% on MT-10-rand under the standard setup of 2 Million environmental steps per task for training. Please refer to more details in the reply to “**Common question on Single-Task SAC**” above.
>
>
>
> **Q1: The paper mentions a "stop-relaunch" trick in [23] but I could not find a reference to that trick in [23].**
>
>
>
> Thanks for pointing this out and sorry that we did not make it clear in the main paper. This “stop-relaunch” trick is contained in the official implementation provided by authors of [23] and is applied to all baselines. We have provided a link pointing to the related code in the Appendix Line 118 (footnote 4). We will make this clear in the main text of the revised paper.
>
>
>
>
>
> **Q2: In line 154, it says " in PaCo, there is a natural way to mitigate this issue without resorting to ad-hoc [23] or more expensive schemes [34]." Isn’t the loss maskout another adhoc trick. In fact this trick also introduces a new hyper-parameter.**
>
>
>
> Thanks for the comment. We agree with you that “ad-hoc” is not an accurate term here. What we meant was that the reset scheme in PaCo is more well blended into the algorithm itself, without interrupting the training compared to the "stop-relaunch" scheme. We will revise the paper to make this clear.
>
>
>
> **Q3: I am not sure I understand why w-reset is used. I understand that we can do it but isn’t PaCo already ignoring the loss when it is greater than epsilon... I didn’t understand the need for both loss masking and w-reset.**
>
>
>
> Ignoring the loss excludes the contribution of the particular exploding loss on parameter updates. With this, we can mitigate the immediate adverse impact of this extreme task loss to other tasks, but could also compromise the overall performance because of the stopped learning on the masked-out tasks (as mentioned in Line 293-295). Since once a loss is masked out, there is no direct gradient for bringing it down. In this case, the learning for that particular task is essentially stopped since its task loss is no longer involved in training.
>
>
>
> Based on this setting, w-reset is used to offer an opportunity to continue learning for the task with exploding loss, by resetting its w-parameter, which will generate a new task parameter vector $\theta_{\rm new}$ when composed with the $\mathbf{\Phi}$ as $\theta_{\rm new}=\mathbf{\Phi} \mathbf{w}_{\rm new}$ without any influence on other non-exploding tasks.
>
>
>
> This offers a new starting point to further learn on the task with previously exploding loss, which naturally further improved over loss-maskout, as shown in Table 2.
>
>
> **Q4: Is the main novelty that they are doing composition in the parameter space ... I just want to make sure I am not missing anything here.**
>
>
>
> Yes, the proposed parameter-compositional form of MTRL method is one of the core contributions in this work. It is simple yet effective and we are not aware of any similar methods published with state-of-the-art performance on standard MetaWorld benchmarks.
>
>
> We regard this simple yet effective approach a valuable contribution to the MTRL community. On top of the proposed compositional form, the separation of task-dependent and task-agnostic parameters also allows us to further design novel schemes like w-reset to improve stability of MTRL training. We admit that the current form has its limitations as discussed in the paper, and we hope this work can serve as a good starting point for further improvement for fellow researchers. We will release the code for reproducing our results with an updated url in the paper to facilitate fellow researchers on future research.

---

> > ### Comment · Reviewer_CCrp · 2022-08-08
> > **Reply to the rebuttal 1/n**
> >
> > Thank you for taking the time to reply.
> >
> > > Weakness: The paper should add the “one-model per task baseline” to their results.
> >
> > >> Thanks for the suggestion. We have now added the one-model per task baseline (Single-Task SAC) as you suggested. Single Task SAC reaches an average success rate of 61.9% on MT-10-rand under the standard setup of 2 Million environmental steps per task for training. Please refer to more details in the reply to “Common question on Single-Task SAC” above.
> >
> > This is a very strange finding. Note that PCGrad, which basically projects gradients to avoid gradient interference, achives 61.7% perf on MT-10-rand and Single-Task SAC gets 61.9%? Even the multi-task SAC, which just uses different exploration coefficient per task, performs better than the one-model per task baseline. I am wondering if the authors tried to tune the baseline properly.
> >
> > > Q1: The paper mentions a "stop-relaunch" trick in [23] but I could not find a reference to that trick in [23].
> >
> > >> Thanks for pointing this out and sorry that we did not make it clear in the main paper. This “stop-relaunch” trick is contained in the official implementation provided by authors of [23] and is applied to all baselines. We have provided a link pointing to the related code in the Appendix Line 118 (footnote 4). We will make this clear in the main text of the revised paper.
> >
> > I checked the line. It says stop the training if the (multi-task) loss becomes larger than 100 million. With such a high loss cutoff, it looks like a "trick" to kill experiments corresponding to bad hyper-parameters and not to account for the increase in the loss corresponding to a single task. Moreover, the "stop-relaunch" trick doesnt make sense as metaworld is a deterministic environment. So "re-launching" the experiment is not going to reduce the loss. Does this make sense or am I missing something here?
> >
> >
> > > Q2: In line 154, it says " in PaCo, there is a natural way to mitigate this issue without resorting to ad-hoc [23] or more expensive schemes [34]." Isn’t the loss maskout another adhoc trick. In fact this trick also introduces a new hyper-parameter.
> >
> > >> Thanks for the comment. We agree with you that “ad-hoc” is not an accurate term here. What we meant was that the reset scheme in PaCo is more well blended into the algorithm itself, without interrupting the training compared to the "stop-relaunch" scheme. We will revise the paper to make this clear.
> >
> > Note the comment about the "stop-relaunch" scheme above.
> >
> > > Q3: I am not sure I understand why w-reset is used. I understand that we can do it but isn’t PaCo already ignoring the loss when it is greater than epsilon... I didn’t understand the need for both loss masking and w-reset.
> >
> > >> Ignoring the loss excludes the contribution of the particular exploding loss on parameter updates. With this, we can mitigate the immediate adverse impact of this extreme task loss to other tasks, but could also compromise the overall performance because of the stopped learning on the masked-out tasks (as mentioned in Line 293-295). Since once a loss is masked out, there is no direct gradient for bringing it down. In this case, the learning for that particular task is essentially stopped since its task loss is no longer involved in training.
> >
> > Thanks for the clarification. Are you ignoring just one gradient update or all subsequent gradient updates on the given task? Subsequent gradient updates should bring it down no?
> >
> > >> Based on this setting, w-reset is used to offer an opportunity to continue learning for the task with exploding loss, by resetting its w-parameter, which will generate a new task parameter vector  when composed with the  as  without any influence on other non-exploding tasks. This offers a new starting point to further learn on the task with previously exploding loss, which naturally further improved over loss-maskout, as shown in Table 2.
> >
> > Noted - thanks!

---

> > ### Comment · Reviewer_CCrp · 2022-08-08
> > **Reply to rebuttal 2 / n**
> >
> > > Q4: Is the main novelty that they are doing composition in the parameter space ... I just want to make sure I am not missing anything here.
> >
> > >> Yes, the proposed parameter-compositional form of MTRL method is one of the core contributions in this work. It is simple yet effective and we are not aware of any similar methods published with state-of-the-art performance on standard MetaWorld benchmarks.. We regard this simple yet effective approach a valuable contribution to the MTRL community. On top of the proposed compositional form, the separation of task-dependent and task-agnostic parameters also allows us to further design novel schemes like w-reset to improve stability of MTRL training. We admit that the current form has its limitations as discussed in the paper, and we hope this work can serve as a good starting point for further improvement for fellow researchers. We will release the code for reproducing our results with an updated url in the paper to facilitate fellow researchers on future research.
> >
> > Noted. Thanks for the clarification. If the main novelty is composition in the parameter space, should baselines like "weight-space ensemble[0]" also be considered? I understand that it is not fair to ask for a new baselines during rebuttal so I wont hold it against the paper but passing it on as it might help to make the paper stronger.
> >
> > [0]: https://proceedings.mlr.press/v9/zhang10c.html
> >
> > > Q5: Computational overhead of different methods.
> >
> > >> For a total of T tasks, PCGrad has a computational complexity of O(T^2), as it needs to compute pairwise task gradient inner products for resolving gradient conflicts. The rest methods have a computational complexity of O(T). The number of network parameters also contributes to the actual computational cost. Methods using simple backbones (e.g., Multi-Head SAC, Multi-Task SAC, FiLM) are usually faster. Other methods have some additional parameters because of their introduced additional modules on top of the same backbone. For example, CARE’s total number of network parameters is affected by the number of encoders in the mixture of encoder networks for processing inputs; SoftModule is affected by the size and number of modules in each layer. PaCo’s total number of network parameters is affected by the size of the parameter set (K).
> >
> > Probably I should have been explicit but I was hoping to see the actual runtime for the algorithms, along with the (say peak) memory consumption.
> >
> > > Q6: The analysis related to Figure 4 is not very convincing … as the same argument applied to "pick-place" while it is quite far in the latent space.
> >
> > >> This is a very good point that we missed mentioning in the paper, and thanks for pointing it out. Actually, it also inspires new ideas for future research.
> >
> > >> Firstly, Figure 4 is the 2D PCA projection of the 10 compositional parameters  for {reach, push, pick-place … } obtained by training PaCo on MT-10-rand, with ~85.4% success rate (Table 1). Actually, among all the 10 tasks, pick-place is a task that has a low success rate after training. Because of this, the  is not a representative  for solving the pick-place task. We conjecture that this is the main reason why it is not close enough to other points with similar behavior patterns in terms of motion trajectories.
> >
> > Noted though I still find the argument (about window-open/window-close/door open/drawer-open. being similar) quite hand-wavy. eg. why is drawer-close and window close far away while drawer open and window open are close. I understand that these are just the PCA representations and it is interesting the certain tasks are closer to each other but I dont find it very convincing.
> >
> > >> Secondly, your question actually inspires us to think further about how to incorporate prior task similarity information (when available) into learning in a general way. In this case, for example, if we have the prior knowledge that "pick-place" is more similar to tasks such as window-open/window-close/door-open/drawer-open than some other task (e.g., button-press top-down), we may have an opportunity to guide the model to learn a better policy for "pick-place". Of course, this is our current intuition only as the inter-skill similarities are typically unavailable in standard MTRL settings. Nevertheless, how to leverage the task similarities when provided or even learn it together is a very interesting direction, and we will explore it in future work.
> >
> > Noted. Thanks
> >
> > I once again thank the authors for taking the time to reply. I still have questions about how well are the single-task baselines trained and about contributions like w-reset which seems to be motivated from "stop-relaunch", which imo is not interepreted correctly. At this time, I am not changing my scores and look forward to further discussion with AC and the other reviewer.

---

> > > ### Author Response · Authors · 2022-08-08
> > > **Clarification responses to Reviewer CCrp (Part 2/2)**
> > >
> > > > Are you ignoring just one gradient update or all subsequent gradient updates on the given task? Subsequent gradient updates should bring it down no?
> > >
> > > The gradient contribution from a task will only be ignored for the current training step if it is higher than the threshold. Therefore, if the loss of that task is still higher than threshold in subsequent training steps (which is typical as explained below), its contribution to the gradient update will still be ignored. If it’s lower, the loss will contribute to gradient update.
> > >
> > > In experiments, subsequent gradient updates won’t bring it down typically, the loss is typically still higher than the threshold, and there is no direct gradient for minimizing the loss of the “masked” task (since the loss of this task is excluded from gradient computation). Therefore, it is common that it will be continually masked out during subsequent training steps.

---

> > > ### Author Response · Authors · 2022-08-08
> > > **Clarification responses to Reviewer CCrp (Part 1/2)**
> > >
> > > **Response to the summary comment.**
> > >
> > > >I still have questions about how well are the single-task baselines trained and about contributions like w-reset which seems to be motivated from "stop-relaunch", which imo is not interepreted correctly. At this time, I am not changing my scores.
> > >
> > > We are happy to help the reviewer to understand any point that you have in question.
> > > In our view, these two points are not technical weaknesses of our method but are due to some mis-understaning on MTRL evaluation protocol and on an empirical trick from an existing codebase.
> > >
> > > We explain these two points below. We do hope that the explanation below can help to resolve the confusions on those two points and we sincerely hope the reviewer can reconsider the rating if these points are clarified.
> > >
> > >
> > >
> > >
> > > **Clarifications:**
> > >
> > > > This is a very strange finding .…. Even the multi-task SAC, which just uses different exploration coefficient per task, performs better than the one-model per task baseline.
> > >
> > > It seems that there is still some confusion on the evaluation protocol and we want to emphasize that it is important to understand the evaluation protocol when interpreting the results.
> > >
> > > **MTRL goal and Protocol.** One of the main goals of MTRL is to train a single policy for solving multiple tasks with fewer training samples, by properly leveraging the samples from other tasks in training. The protocol is designed to better reflect this *sample efficiency improvement* objective.
> > >
> > > In MetaWorld,
> > >   - **5 Million env steps/task is required for solving each task individually**. The tasks in Meta-World are designed such that they can be solved with standard single task RL algorithms (e.g. single-task SAC), given enough environmental steps (varies for different tasks, for harder tasks, it is about *5 Million/task*), single-task SAC can converge to ~95% success rate on MetaWorld.
> > >
> > >    - **2 Million env steps/task as a protocol for comparing sample efficiency**. Previously established protocol used 2M env steps for training MTRL methods since it can better highlight the sample efficiency improvements. Given the same number (2M) of env steps that is reasonably far from the saturation point (5M), a method that has a higher sample efficiency will have a higher success rate.
> > >
> > > Once this is clear, under the standard 2M/task setting, it is natural to observe some sample efficiency improvement of multi-task SAC (which shares all parameters apart from temperatures) compared with single-task SAC, which has no parameter sharing at all (a setting that is least sample efficient, although the performance bound could be higher in the unlimited env steps setting, e.g. ~95% success rate for 5M/task).
> > >
> > > > I checked the line. It says stop the training if the (multi-task) loss becomes larger than 100 million. With such a high loss cutoff, it looks like a "trick" to kill experiments corresponding to bad hyper-parameters and not to account for the increase in the loss corresponding to a single task.
> > >
> > > We want to clarify that this is not purely an interpretation but what actually happens in practice. We used this codebase’s implementation for all the base methods reported and we’ve actually experienced “stop trick” being triggered several times in 10 random seeds experiments (e.g. multi-head and multi-task SAC encountered this more frequently).
> > >
> > > Therefore it’s not only a trick for hyper-parameter tuning, but also a way to stop training on seeds that “fails” in the middle of training in experiments. With this trick applied, the “stopped” ones are excluded from the results reported (and additional new ones have to be trained in addition). This might be a reason that Multi-Task SAC has relatively high performance compared to PCGrad (PCGrad is more stable and never triggers the stop in training experiments).
> > >
> > > We admit that by using this trick, methods that trigger this more (e.g. Multi-Task SAC compared to PCGrad) will have more advantage in the performance, since the number of samples used in the stopped trains are not counted in.
> > > This was inherited from previous work (CARE codebase).
> > > That’s actually one of the motivations that we want to introduce the PaCo method which offers a natural in-algorithm stabilization scheme.
> > >
> > >
> > > > Moreover, the "stop-relaunch" trick doesnt make sense as metaworld is a deterministic environment. So "re-launching" the experiment is not going to reduce the loss. Does this make sense or am I missing something here?
> > >
> > > The re-launched training will use a new seed that is different from before. With a new seed, NNs are initialized differently (and also random goals are re-initialized in MT10-rand). This could lead to different learning dynamics. In “w-reset” we don’t need to re-launch (reset all parameters) but can simply reset the networks corresponding to the extreme task loss only, and without interrupting the training process.

---

> > > > ### Comment · Reviewer_CCrp · 2022-08-09
> > > > **Reply v2 1/n**
> > > >
> > > > > One of the main goals of MTRL is to train a single policy for solving multiple tasks with fewer training samples, by properly leveraging the samples from other tasks in training. The protocol is designed to better reflect this sample efficiency improvement objective.
> > > >
> > > > Noted
> > > >
> > > > > In MetaWorld, 5 Million env steps/task is required for solving each task individually. The tasks in Meta-World are designed such that they can be solved with standard single task RL algorithms (e.g. single-task SAC), given enough environmental steps (varies for different tasks, for harder tasks, it is about 5 Million/task), single-task SAC can converge to ~95% success rate on MetaWorld.
> > > >
> > > > Noted
> > > >
> > > > > 2 Million env steps/task as a protocol for comparing sample efficiency. Previously established protocol used 2M env steps for training MTRL methods since it can better highlight the sample efficiency improvements.
> > > >
> > > > This is not correct. PCGrad, SoftModularization and CARe are proposed in context of MetaWorld V1 where 2M steps are shown to be sufficient for the single task baseline to solve the task. so the 2M steps used by those papers is not an arbitrary choice. I think some of these papers also considered 100K variants (which are fine to include imo) but the core results are on 2M. In the current case, the Metaworld version needs 5M steps for the single task baseline. So the reasonable set up would be to train the algos on 5M steps (to ensure the approaches had sufficient steps to converge). The authors are ofc free to show results at intermediate steps as well (which is useful imo but not sufficient).
> > > >
> > > > I want to be clear here - my complain is not why single task baseline better or worst. Single task baseline being better would neither "surprise" me nor "disappoint" me. I was indeed surprised to see the baseline under-perform, which the authors explained could likely be because of underfitting. Now the concern I have is, how do we know the other algos are not underfitting? Reporting all results till 5M steps would address this.
> > > >
> > > > > Once this is clear, under the standard 2M/task setting, it is natural to observe some sample efficiency improvement of multi-task SAC (which shares all parameters apart from temperatures) compared with single-task SAC, which has no parameter sharing at all
> > > >
> > > > Not necessarily. " it is natural to observe some sample efficiency improvement of multi-task SAC (which shares all parameters apart from temperatures)" -> if this were to be true, we would not encounter the problem of "gradient interference" in multi-task learning (which PCGrad is used to fix).
> > > >
> > > > >  single-task SAC, which has no parameter sharing at all (a setting that is least sample efficient,
> > > >
> > > > This is also not necessarily true - There are plenty of cases where training one-policy-per-task is not the "least sample efficient" choice including MetaWorld v1.
> > > >
> > > > Also note that the argument about PCGrad vs Single Task baselines is not addressed.

---

> > > > > ### Comment · Reviewer_CCrp · 2022-08-09
> > > > > **Reply v2 2/n**
> > > > >
> > > > > > We want to clarify that this is not purely an interpretation but what actually happens in practice. We used this codebase’s implementation for all the base methods reported and we’ve actually experienced “stop trick” being triggered several times in 10 random seeds experiments (e.g. multi-head and multi-task SAC encountered this more frequently).
> > > > >
> > > > > While I have no way of ascertaining what is the correct "interpretation", I do have an issue with the experiment protocol in the current paper. If the authors feel that the “stop trick” is added to help the baselines, they should report the results without the trick - something along the lines of "we note that about 10% of the experiments crash. For the sake of comparison, we ran experiments with additional seeds", instead of using a trick that is not in the baseline papers.
> > > > >
> > > > > > Therefore it’s not only a trick for hyper-parameter tuning, but also a way to stop training on seeds that “fails” in the middle of training in experiments. With this trick applied, the “stopped” ones are excluded from the results reported (and additional new ones have to be trained in addition). This might be a reason that Multi-Task SAC has relatively high performance compared to PCGrad (PCGrad is more stable and never triggers the stop in training experiments).
> > > > >
> > > > > As noted in the previous comment, this is problematic. I understand that rewording this does not change the results for the authors algorithms so I have no complaints on that part. My complaint is, introducing the "stop trick" is problematic because the future papers would refer the current paper (or the baseline papers) to justify the use of this "trick" and not all reviewers would read the baseline papers. If you think the trick is not in the algorithm then dont use it and simply report the % of crashes. Dont invent a "trick" that other future work could latch onto. For example, a future work would set the 100M threshold to say 10 and discard potentially bad run under this disguise. My understanding is, at 100M loss, the network is going to Nan anyways so it doesnt change any of the results for the authors but the underlying problem (that the author's work fixes) is the crashing on the model. Introducing the "trick" distracts from that
> > > > >
> > > > >
> > > > > > The re-launched training will use a new seed that is different from before. With a new seed, NNs are initialized differently (and also random goals are re-initialized in MT10-rand).
> > > > >
> > > > > Noted - Also refer to reply above.
> > > > >
> > > > > > The gradient contribution from a task will only be ignored for the current training step if it is higher than the threshold. Therefore, if the loss of that task is still higher than threshold in subsequent training steps (which is typical as explained below), its contribution to the gradient update will still be ignored. If it’s lower, the loss will contribute to gradient update. In experiments, subsequent gradient updates won’t bring it down typically, the loss is typically still higher than the threshold, and there is no direct gradient for minimizing the loss of the “masked” task (since the loss of this task is excluded from gradient computation). Therefore, it is common that it will be continually masked out during subsequent training steps.
> > > > >
> > > > > Again, this is confusing. You see one step of high loss, and you skip that loss. Good so far. Then the argument is, "subsequent gradient updates won’t bring it down typically". Is the "it" referring to the loss in the previous step? But that was ignored.

---

> > > > > ### Author Response · Authors · 2022-08-09
> > > > > **Responses to further comments**
> > > > >
> > > > > **Evaluation protocols**
> > > > >
> > > > > We thank reviewer CCrp for the valuable feedback and discussions on evaluation protocols. In our work, we put the most effort into improving the performance and stability of the method and put less effort into evaluation. We do agree that how to evaluate MTRL methods is very important. Both the performance on solving tasks (success rate a single model can reach across multiple tasks) and sample efficiency (number of samples required to reach reasonable performance) should be considered for a good evaluation metric. The current evaluation method (2M/task) is designed to reflect this trade-off.
> > > > >
> > > > > Some example training curves of single-task SAC of different runs for some of the tasks that require more env steps are shown here for reference (door-open: https://imgur.com/4XFV1bh    peg-insert-side: https://imgur.com/a/7J33mVR).
> > > > > For the previously mentioned 5M number, we took a pessimistic perspective regarding variations in different runs as shown above (e.g., considering the steps required in the worst case).
> > > > > However, from an optimistic perspective,  it can be observed that for single-task SAC, at 2M, some seeds could already lead to non-trivial learning (e.g., the policy with reasonable performance). Therefore evaluating at 2M is still a valuable setting that is non-trivial on the one hand while could also highlight the sample efficiency on the other hand. (Under this setting, Multi-Task SAC already reaches its performance limitation, but there are chances that PCGrad is under-perform and still has space to improve with more steps)
> > > > >
> > > > > **“stop-relaunch” trick**
> > > > >
> > > > > We also thank the reviewer for your further explanation on the “stop-relaunch” trick. Now we have understood your comments on this point better: which is the potential for future papers to justify the use of this "trick" by referring to this work. While our original intention for adopting the trick following CARE repo was to make the baselines stronger, we admit it could bring some confusions as you mentioned. We will update the results removing this trick and report the % of crashes as you suggested in revision if that helps in resolving this issue. Please note this would only decrease the performances of some baseline methods (mainly Multi-Task SAC, and Multi-Head SAC) but has no impact on the proposed method.
> > > > >
> > > > > **Clarification on gradient updates.**
> > > > >
> > > > > > Then the argument is, "subsequent gradient updates won’t bring it down typically". Is the "it" referring to the loss in the previous step? But that was ignored.
> > > > >
> > > > > ”It“ refers to the loss of the task-$i$ computed at the current training step, where $i$ corresponds to the task with exploding loss in the previous step (”it“ corresponds to $L_1^{t}$ in the example below).
> > > > >
> > > > > For example, for the cases of two task losses $L=L_1 + L_2$, at training step $t-1$, one task has an exploding loss (say task 1, meaning $L_1^{t-1} > \eta$). At step $t-1$, $L_1^{t-1}$ is ignored and only gradient of $L_2^{t-1}$ is used ($L^{t-1}=0*L_1^{t-1}+L_2^{t-1}$). Since the effective gradients at time $t-1$ are only from task 2, parameters related to task 1 are unlikely to be updated in the direction aligned with reducing $L_1$.
> > > > >
> > > > > At training step $t$, the same procedure is carried out: both $L_1^t, L_2^t$ are first calculated and compared with the threshold $\eta$. In most cases $L_1^t$ will be masked out again since typically the task loss remains high ($L_1^t>\eta$) without direct gradient update from the corresponding task.

---

### Author Response · Authors · 2022-08-02
**Thanks for the reviews and responses to common questions**

​​We thank all the reviewers (**R1**-CCrp, **R2**-DW33, **R3**-U1FU) for their reviews. The reviewers appreciated the:



1. novelty of the method (**R2**-DW33, **R3**-U1FU),

2. strong results (**R1**-CCrp, **R2**-DW33, **R3**-U1FU),

3. interestingness (**R2**-DW33, **R3**-U1FU),

4. good presentation (**R1**-CCrp, **R2**-DW33, **R3**-U1FU).



We first address a common question and then reply to each individual question.

To avoid confusion on some explicit pointers (e.g. to line number, page number etc) and facilitate review, we did not update the pdf and instead post all the relevant information in responses below. The paper will be revised based on the valuable comments from all reviewers and incorporate the additional results posted here.





**Common question on Single-Task SAC**: (**R1**-CCrp: “one-model-per-task baseline”; **R2**-DW33: “a baseline where each task has its own policy”; **R3**-U1FU: “the performance of single-task model where you use separate network for each task”)



- This is a valuable point, and we apologize that we’ve inappropriately assumed the knowledge on some of the established results in MTRL literature and did not make this clear enough in the paper. We will revise the paper and add more information on it.

- The tasks in Meta-World are designed such that they can be trained with standard single task RL algorithms (e.g. SAC). Given enough environmental steps (varies for different tasks, for harder tasks it is about 5 Million/task), they can converge to close to 1.0 success rate. Since hard tasks may fail for some seed, the average success rate at convergence for the 10 tasks in MT10 is ~ 95.0% (each SAC policy is trained with 5Million environmental steps per task). Therefore, in previous works (e.g., CARE), Single-Task SAC performance under this setting is regarded as an “upper bound”.

- One of the main goals of MTRL is to train a single policy for solving multiple tasks with less training samples by leveraging proper forms of parameter sharing. To compare different methods from this perspective, the standard evaluation protocol (e.g. in CARE) is to train different methods for the same number of environmental steps (2M/task for meta-world) and then compare their performances. Following reviewers’ suggestions and the standard evaluation protocol, we add a new experiment for a 2M env steps/task for Single-Task SAC training, which results in a 61.9% average success rate. Single-Task SAC cannot reach its convergence performance for all tasks if limited to 2M per task for training. We will add this as a baseline and make it more clear in the revision.

---

### Meta-Review · Area_Chair_zMAo · 2022-08-25

**Recommendation:** Accept
**Confidence:** Less certain

**Metareview:**

This paper proposes a multi-task RL architecture that composes task-independent parameters and task-dependent parameters to construct a task-specific policy. The results on the MetaWorld multi-task learning benchmark shows that the proposed method outperforms relevant baselines. In general, the reviewers found the idea interesting and novel, and the results are solid. Several concerns about the lack of single-task baseline and the lack of scores given intermediate steps were addressed with updated results during the rebuttal period, which made all the reviewers lean towards acceptance. Thus, I recommend accepting this paper. In the meantime, there are still remaining comments about the result beyond 2M steps (suggested by Reviewer CCrp) and additional baselines (suggested by Reviewer DW33). I highly suggest the authors to include these results for the camera-ready version.

**Award:**

No

---

### Decision · Program_Chairs · 2022-09-14

Accept